# Robust hemostatic bandages based on nanoclay electrospun membranes

Yan Cui [1,2,7], Zongwang Huang [1,7], Li Lei[3,7], Qinglin Li[4,5,7], Jinlong Jiang[6], Qinghai Zeng[3,8], Aidong Tang[2,8], Huaming Yang [1,8] & Yi Zhang [1,8 ✉]

Death from acute hemorrhage is a major problem in military conflicts, traffic accidents, and surgical procedures, et al. Achieving rapid effective hemostasis for pre-hospital care is essential to save lives in massive bleeding. An ideal hemostasis material should have those features such as safe, efficient, convenient, economical, which remains challenging and most of them cannot be achieved at the same time. In this work, we report a rapid effective nanoclay-based hemostatic membranes with nanoclay particles incorporate into poly-vinylpyrrolidone (PVP) electrospun fibers. The nanoclay electrospun membrane (NEM) with 60 wt% kaolinite (KEM1.5) shows better and faster hemostatic performance in vitro and in vivo with good biocompatibility compared with most other NEMs and clay-based hemo-stats, benefiting from its enriched hemostatic functional sites, robust fluffy framework, and hydrophilic surface. The robust hemostatic bandages based on nanoclay electrospun mem-brane is an effective candidate hemostat in practical application.

[1] Department of Inorganic Materials, School of Minerals Processing and Bioengineering, Central South University, 410083 Changsha, China. [2] College of Chemistry and Chemical Engineering, Central South University, 410083 Changsha, China. [3] Department of Dermatology, the Third Xiangya Hospital, Central South University, 410013 Changsha, China. [4] Cancer Hospital of the University of Chinese Academy of Sciences (Zhejiang Cancer Hospital), Hangzhou, China. [5] Institute of Cancer and Basic Medicine (IBMC), Chinese Academy of Sciences, Hangzhou, China. [6] Jiangsu Provincial Key Laboratory of Palygorskite Science and Applied Technology, Huaiyin Institute of Technology, 223003 Huaian, China. [7] These authors contributed equally: Yan Cui, Zongwang Huang, Li Lei, Qinglin Li. [8] These authors jointly supervised this work: Qinghai Zeng, Aidong Tang, Huaming Yang, Yi Zhang. ✉email: yee_z10@csu.edu.cn

Acute hemorrhage is one of the greatest causes of death each year around the world[1]. A practical prehospital care solution for acute hemorrhage control, especially in the case of warfare, catastrophes and accidents, is urgently needed and is characterized by high clinical demand and large market potential[2,3]. Rapid effective hemostatic material application in initial hemorrhage phases can lead to an extended rescue time, resulting in a decline in high mortality rates due to excessive bleeding[2]. However, it is still challenging for most hemostatic materials to quickly and safely control hemorrhage from severe bleeding wounds or cuts[1,4]. The desired hemostatic materials must be designed and developed to achieve rapid definitive hemostasis for both noncompressible (visceral organs) and compressible (cervical, trunk and head regions near the heart and brain) massive bleeding, both of which involve fatal bleeding and are causes of equal concern[4–6].

To date, various hemostasis products have been reported, including injectable glues (gelatin[4]), hemostatic bandages (polymer composites[7] and cellulose fabrics[8]), and procoagulant powders[9] (siliceous oxides[10] and inorganic clays[11,12]). Zeolites and clays, the main components of Z-Medica products (Clotting Sponges, Clotting Gauze (ClG) and Combat Gauze (CoG), are widely accepted as effective hemostats[6,13]. Clays can accelerate the stoppage of bleeding by cooperating with physical hemostatic functions, which facilitate plasma absorption to increase blood cell concentration, and with endogenous hemostasis pathways for negatively charge-stimulated clotting factors[13–15]. Generally, most traditional clay hemostatic materials are in the form of a powder (requiring dressing with gauze) or a clay powder-impregnated gauze (easily detached due to poor adhesion)[6,16]. Although clays have shown effective hemostatic performance, prolonging the rescue time, the application of clay particles to bleeding wounds may cause inflammation in surrounding tissues and distal thrombosis in vivo[13,17]. Therefore, it is important to develop membranes or other alternative materials to replace these powder-based materials.

Hemostatic clay particles can be connected by polymers to transform clay powders into clay membranes for acute hemostasis. A few types of biomaterials using nanoclays have been developed as effective hemostats, including hydrogels (e.g., gelatin/polyacrylamide (PAAm)/laponite hydrogels[18]) and sponges (e.g., graphene-kaolinite or graphene-montmorillonite composite sponge[14,19]), which are strict and complicated treatments with high costs as medical hemostats. Hydrogels and sponges show good shape memory but poor mechanical strength due to their inherent interconnected macroporous structures, and polymer binders fill and block the clay pores and layer gaps, causing low clay utilization as active components[6,16]. Electrostatic spinning is one of the most commonly used membrane synthesis methods; this method can be incorporated with nanoclays and enables the easy preparation of light, fluffy, and soft membrane materials[20–22]. Pure polymer electrospun membranes will sharply spontaneously shrink within a short time after preparation[23], similar to the contraction of cell membranes, which may have potential applications in emerging technologies (flexible electrodes[24] or stretchable materials[25,26]); however, reducing the effective area of the membrane will increase costs and decrease hemostasis capability. Thus, an ideal design for clay hemostatic membranes should be developed with the characteristics of safety, rapid hemostasis, high active component utilization and a robust antishrinking framework[27–29].

In this work, we aim to develop a nanoclay electrospun membranes (NEMs) exhibiting a superior hemostatic capability in vitro and in vivo, benefiting from enriched hemostatic functional sites, a robust fluffy framework and a hydrophilic surface. Sheet-like kaolinite, tube-like halloysite and rod-like palygorskite have been chosen as the representative layered aluminosilicate structures, incorporating into polyvinylpyrrolidone (PVP) and generating the biocompatible electrospun frameworks. The PVP fibers provide a robust framework for the clay particles, while the nanoclay can be dispersed into the PVPEM three-dimensional (3D) network as a filler to support the membrane framework, effectively resisting spontaneous shrinkage. Thus, NEMs exhibit great potential for use as rapid robust hemostats that may be applied as compressible hemorrhage control devices.

## Results and discussion

**Internal interactions in the nanoclay electrospun membranes.** Rapid robust bandages based on NEMs were prepared by electrospinning with sheet-like kaolinite[27], tube-like halloysite[28] and rod-like palygorskite[29] (Fig. 1a and Supplementary Fig. 1). The nanoclays were incorporated with PVP electrospun membranes (PVPEM), namely (Supplementary Table 1), kaolinite electrospun membranes (KEM$_{1.5,2.0,2.4}$), halloysite electrospun membranes (HEM$_{1.5}$) and palygorskite electrospun membranes (PEM$_{1.5}$), respectively. The high flexibility (easily twisting 360° in situ) and manufacturability of NEMs are perfectly inherited from the electrospun substrate (Fig. 1b), making them suitable for hemostatic applications[4,8]. The scanning electron microscope (SEM) images (Fig. 1c) showed that kaolinite particles with individual sheet physical structures were well-dispersed and partly exposed on the KEM surface (the best structures were obtained with a mass ratio of 1:1.5 (KEM$_{1.5}$). However, bulky agglomerations on KEM$_{2.0}$ and KEM$_{2.4}$ were attributed to the overhigh kaolinite particle mass ratio, breaking the perfectly incorporated structure and restricting its hemostatic efficiency[21]. The tube-like halloysite and rod-like palygorskite physical structures were disadvantageous to exposure at the membrane surface, and all materials were wrapped in fibers, causing slight agglomerations on HEM$_{1.5}$ and PEM$_{1.5}$.

Fourier transform infrared (FT-IR) spectra and thermogravimetry-differential scanning calorimetry (TG-DSC) curves were used to characterize the synthesis and internal bonding of the NEMs. Bands at ~3696–3615 cm$^{-1}$, 1100–1000 cm$^{-1}$, 950−900 cm$^{-1}$, 800–750 cm$^{-1}$ and 696–680 cm$^{-1}$ were assigned to Al-OH stretching vibrations, skeleton Si-O stretching vibrations, Al-OH bending vibrations, Si-O-Al vibrations and OH vibrations (Supplementary Fig. 2a), respectively, in agreement with the characteristic peaks of kaolinite, halloysite, and palygorskite[30–32]. The bands at ~2955 cm$^{-1}$, 1680 cm$^{-1}$, 1440 cm$^{-1}$ and 1290 cm$^{-1}$ were assigned to C-H stretching vibrations, C=O stretching vibrations, C-H bending vibrations and C-N stretching vibrations (Supplementary Fig. 2a), respectively[33], in agreement with the characteristic peaks of PVP. The C=O stretching vibration peak shift and hydroxyl stretching vibration peak were from 1659 to 1671 cm$^{-1}$ and at ~3620 cm$^{-1}$, respectively, as visible in the enlarged image (Fig. 1d), which might be due to hydrogen bonds (O-H…O) and (O-H…N) at the clay/PVP interface. The shift increase was concomitant with increased hydrogen bonding interactions in the material. The magnitude of the shift indicates that kaolinite has a stronger ability than halloysite and palygorskite to form hydrogen bonds with PVP. This phenomenon is related to the crystal structures of the three clay materials: kaolinite has more exposed hydroxyl groups (including interlayer and surface defect Al-OH) than do tube-like halloysite and rod-like palygorskite. According to TG-DSC thermal analysis (Supplementary Fig. 2b and Supplementary Discussion), severe shrinkage of the PVP fibers and blockage of the membrane structure were effectively resisted due to the strong interaction between clay particles and PVP fibers, thus supporting the electrospun framework structures.

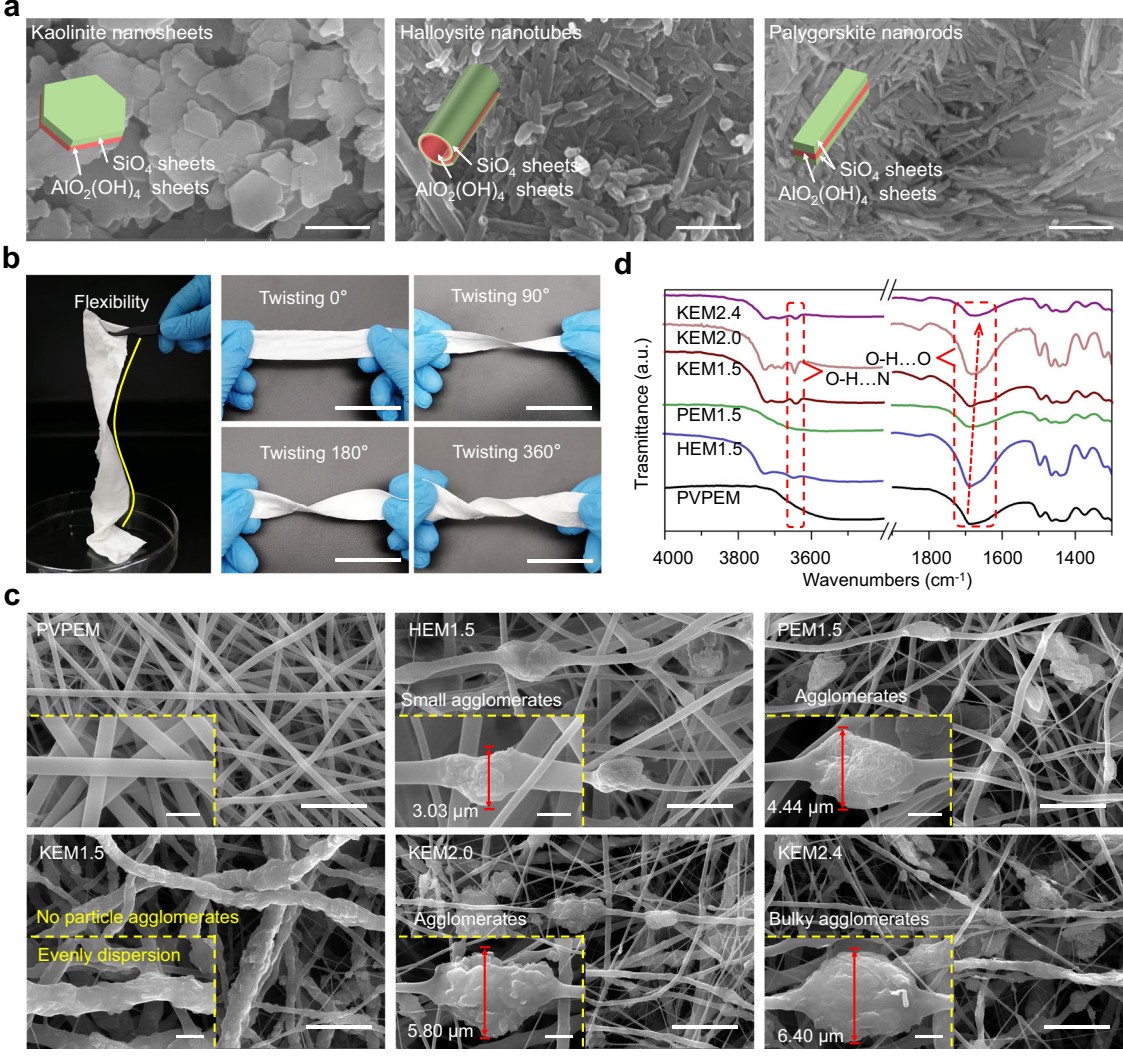

**Fig. 1 Fabrication, surface morphology and characteristics of the nanoclay membranes. a** SEM images of kaolinite, halloysite and palygorskite, respectively. Scale bar, 500 nm. **b** Photographs of clay-membrane strips with different extents of in situ twisting. Scale bar, 10 cm. **c** SEM images of PVPEM, HEM$_{1.5}$, PEM$_{1.5}$, KEM$_{1.5}$, KEM$_{2.0}$, and KEM$_{2.4}$, respectively. Scale bar, 10 μm. The insets in (**c**) are the corresponding magnified SEM images. Scale bars in the insets of (**c**), 2 μm. **d** The localized magnified FT-IR spectra of PVPEM, HEM$_{1.5}$, PEM$_{1.5}$, KEM$_{1.5}$, KEM$_{2.0}$ and KEM$_{2.4}$, respectively. Data were analyzed from at least three independently experiments in (**a**) and (**c**). Source data are provided as a Source Data file.

**Self-supported three-dimensional nanoclay-organic frameworks**. The NEM framework was investigated by observing the surface morphology and structural characteristics. KEM showed fluffier 3D network structures than PVPEM, as verified by the distances between NEM layers (KEM$_{1.5}$, 412.61 μm; KEM$_{2.0}$, 270.89 μm; KEM$_{2.4}$, 374,87 μm; PVPEM, 214.82 μm) under the same conditions, which indicated a high water uptake capability for absorbing wound exudate to avoid bacterial infection and facilitate healing[12,16] (Fig. 2a). PVPEM completely lost the 3D network structure because of spontaneous shrinkage upon deposition in air (the inset of Fig. 2a), resulting in a decrease in hydrophilicity, which is harmful to the long-term use of hemostatic materials[34]. Well-dispersed and partly exposed kaolinite particles on the KEM surface provided functional hemostatic sites and improved the hydrophilicity of the material (water contact angle from 100° to 62°, Supplementary Fig. 2c and Supplementary Discussion), which is related to the regulation of surface properties by intermolecular hydrogen bond interactions (Fig. 2b). HEM$_{1.5}$ and PEM$_{1.5}$ had 3D structures and hydrophilic surfaces, but sadly, particle agglomerates and poor surface exposure hampered hemostatic application[28,29] (Figs. 1c, 2a).

Single fibers in the NEMs were investigated by SEM, transmission electron microscope (TEM) and energy dispersive X-ray spectroscopy (EDX) element analysis to assess the nanoclay particle dispersion and inner structure characteristics. Most kaolinite particles were totally trapped in the fibers with large and evenly diameters (especially KEM$_{1.5}$, Supplementary Fig. 3 and Supplementary Discussion), and partly exposed to the fiber surface due to their individual sheet-like structures (Fig. 2c, d); this configuration was crucial to blood contact and accelerated coagulation. The polymer layer became thinner with increasing kaolinite particle ratio (Fig. 2d, e), resulting in weak adhesion interactions and thus increasing the risk of thrombus from particle exfoliation[6]. The tube-like halloysite and rod-like palygorskite were trapped in the radial direction along the fibers and rarely exposed to the surface (shown clearly in Fig. 2c, d), restricting blood contact. Furthermore, the element analysis results (Supplementary Fig. 4 and Supplementary Discussion) match well with the distribution and structure characteristics from SEM and TEM analysis, indicating excellent hemostasis potential for KEM (best: KEM$_{1.5}$) due to its integrated structure and self-supported 3D framework.

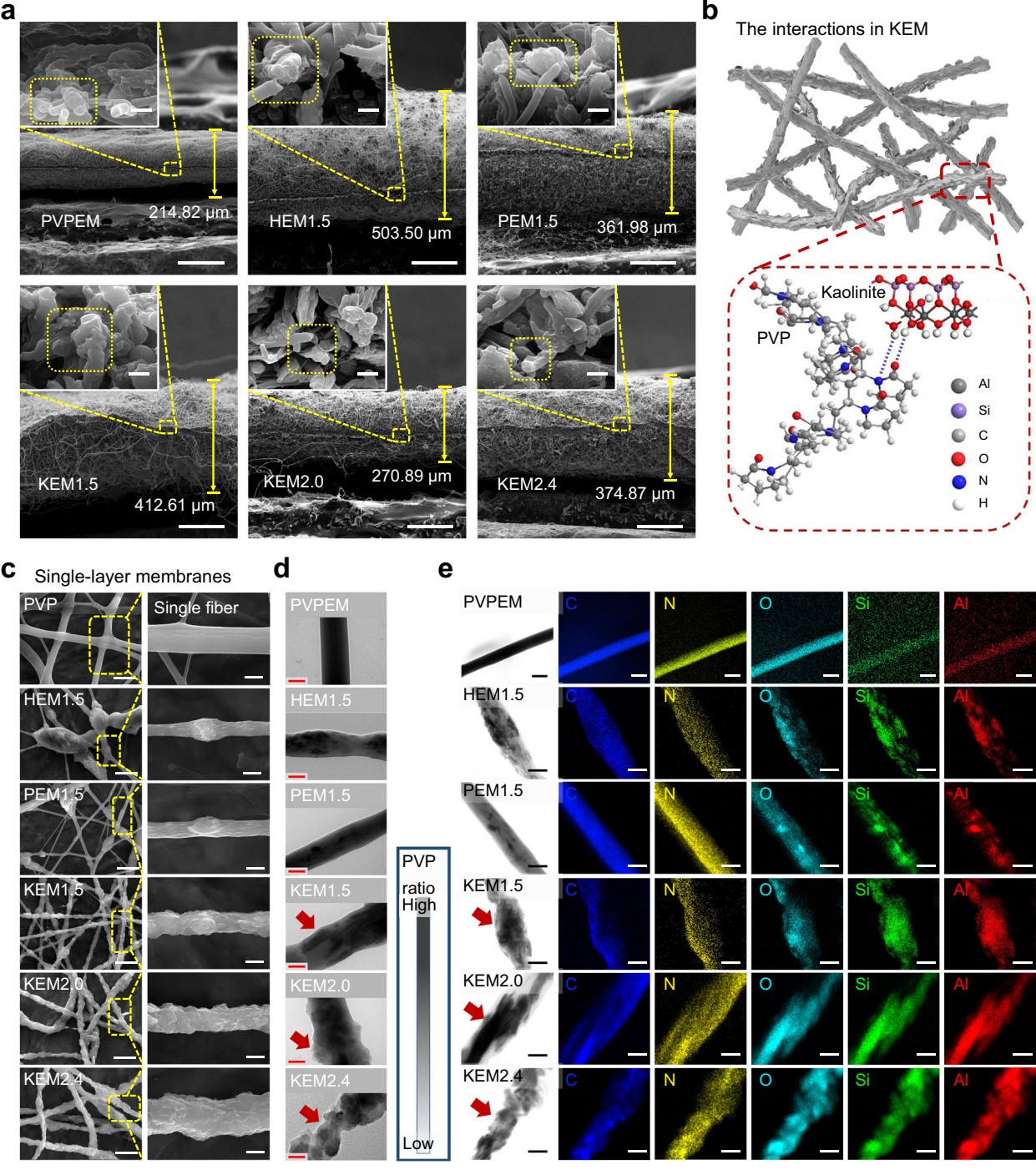

**Fig. 2 The inner morphology, interactions and dispersion characteristics for nanoclay membranes. a** Cross-section SEM images of PVPEM, HEM$_{1.5}$, PEM$_{1.5}$, KEM$_{1.5}$, KEM$_{2.0}$, and KEM$_{2.4}$. Scale bar, 2 μm. The inset in (**a**) are the corresponding magnified SEM images of fiber section. Scale bars in the insets of (**a**), 2 μm. **b** The illustration of the bonding interaction between kaolinite and PVP. **c** The SEM images of single-layer membranes (including the single fiber in the inset), (**d**) TEM and (**e**) mapping analysis images for PVPEM, HEM$_{1.5}$, PEM$_{1.5}$, KEM$_{1.5}$, KEM$_{2.0}$, and KEM$_{2.4}$, respectively. Scale bars in (**c**–**e**) were 10 μm, 0.5 μm and 1 μm, respectively. Scale bars in the insets of (**c**), 2 μm. Data were analyzed from at least three independently experiments in (**a**) and (**c**–**e**). Source data are provided as a Source Data file.

**Shrinkage and in vitro coagulation properties.** The results of the shrinkage experiment and zeta potential values were used to further characterize the interactions in the nanoclay-organic frameworks. The zeta potential values of palygorskite, halloysite, and kaolinite (Supplementary Fig. 5a) showed positive growth after incorporation of PVP, ranging from −16.6 ± 0.3 mV to −1.6 ± 0.1 mV, −15.0 ± 0.3 mV to −2.4 ± 0.1 mV, and −16.6 ± 0.5 mV to −2.3 ± 0.1 mV, respectively, indicating the generation of electrostatic interactions between the PVP and nanoclay. At constant temperature and humidity, the NEM frameworks could effectively resist shrinkage (best for KEM$_{1.5}$, <30% shrinkage ratio), while PVPEM shrank sharply (>75% shrinkage ratio) after the 7-day shrinkage test (Supplementary Fig. 5b and Supplementary Discussion), which was similar to the contraction of cell membranes for preservation[23]. The inherent spontaneous shrinkage might be related to solvent evaporation and the

degradation for PVPEM[34,35]. Importantly, the NEMs were well retained without degradation, while PVPEM was almost completely degraded after 2 days of exposure to air (Fig. 3a). The good resistance to shrinkage and degradation benefited from synergy between the PVP fibers and kaolinite: The kaolinite nanosheets actively stabilize the framework and regulate the spontaneous shrinkage and degradation of PVP fibers through intermolecular interactions with the PVP fibers[23], and the PVP fibers supply a robust framework for the kaolinite particles[34–36].

The NEMs with high permeability and flexible texture fitted perfectly on the skin (Fig. 3b), meeting the 'standard of care' level for commercial products, indicating good breathability for accelerating wound healing and high comfort properties that are suitable for wound application under various traumatic conditions[8]. The hemostasis capability of the NEM frameworks was investigated by an in vitro blood coagulation test and assessment of the blood absorption ability[5] (Fig. 3c and Supplementary Fig. 6). The observational results clearly showed that KEM and the commercial controls (CoG and ClG) could facilitate blood permeation and coagulation within 2–5 min (best for KEM$_{1.5}$, <2 min), while blood was difficult to absorb in PVPEM (control), HEM$_{1.5}$, and PEM$_{1.5}$ within 5 min (Supplementary Fig. 6 and Supplementary Discussion). KEM$_{1.5}$ showed fewer unbound blood components than did the other samples (Fig. 3c), which is a quality criterion of blood clotting indicating low hemolysis.

The coagulation capability and status of the clot were investigated by continuous experimental observation and evaluation of the absorbance (optical density, OD) of the supernatant[5,16]. The clot after NEM treatment showed a steady state (best: KEM$_{1.5}$, almost no hemolysis) after 120 min, while the clot with PVPEM showed gradual hemolysis, and almost the whole clot was finally hemolytic (Fig. 3c), indicating poor coagulation capability in vitro without active components. The supernatant relative absorbance (OD) value of the KEM$_{1.5}$ sample was considerably lower than those of the other samples (less than one of twelfth times that of PVPEM), indicating that KEM (best: KEM$_{1.5}$) could effectively accelerate blood coagulation in vitro.

The KEM coagulation mechanism was further analyzed by SEM images, Flow Cytometry (FCM) and Thromboelastogram. Fig 3d (surface) and Supplementary Fig. 7 (interior) showed that KEM$_{1.5}$ with 2D multilayered structure and 3D reticular architecture facilitated to restrict blood flow and capture blood components. In FCM analysis (Supplementary Fig. 8)[37,38], the percentage of CD61$^+$ cells in KEM$_{1.5}$ group (0.49%), ClG group (2.92%), CoG group (2.55%) showed an obvious decrease compared to blank group (93.48%), due to the free platelet counts decrease caused by hemostats capture[37,39,40]. Importantly, the percentage of CD61$^+$/CD62P$^+$ cells in KEM$_{1.5}$ group (80.28%) was highest among all groups (blank: 24.68%; ClG: 74.98%; CoG: 71.43%), indicating that KEM$_{1.5}$ have an excellent platelet activation effect[39].

In Thrombelastograph Thromboelastogram (TEG) analysis (Supplementary Fig. 9)[41–43], KEM$_{1.5}$ group (Coagulation composite index CI: 1.30; Reaction time R: 1.6 min) exhibited a hypercoagulability with high clotting factors activity compared to blank group (CI: −8.07; R: 11.80 min). As previous work and the related literatures[13,19,42], the whole blood contacts with the kaolinite particles will activate kininogen, plasma kallikrein, and factor XII to trigger the process of coagulation. All the values of Maximum amplitude (MA), Clot formation time (K), Angle, Peripheral blood platelets-monocyte aggregates, Prediction of fibrinolysis index (EPL), Percentage lysis 30 min post-MA (LY30) and Mechanical strength of clot (G) were among the normal ranges in KEM$_{1.5}$ group, maintaining a general-level functions on platelets, fibrinogen (FIB), fibrinolysis and clot strength. Thus, a full and easy contact with the active particles on KEM$_{1.5}$

can accelerate the blood components aggregation (illustration in Fig. 3e) and the contact activation (intrinsic) process of clotting (including factor XII, thrombin and fibrin formation until the final clot, illustration in Supplementary Fig. 10).

**In vivo hemostatic performance**. The emergency hemostasis performance of the NEMs was further investigated by an in vivo rat tail amputation model and a rat liver and spleen hemostasis model, which assessed acute bleeding events in the trunk and visceral organs[13,16]. When active bleeding wounds (the initial bleeding was assessed to ensure a same test condition, Supplementary Fig. 11) of the liver and spleen occurred, the bleeding time was more than 300 s without hemostatic material treatment. It showed a long hemostatic time for the control group (common medical gauze: liver, 164 ± 56 s; spleen, 201 ± 52 s, Supplementary Fig. 12) and PVPEM group (liver: 144 ± 22 s, spleen: 227 ± 30 s). The bleeding times showed clear decreases with NEM treatment: KEM$_{1.5}$ (liver: 86 ± 10 s, spleen: 102 ± 16 s), KEM$_{2.0}$ (liver: 108 ± 15 s, spleen: 118 ± 16 s) and KEM$_{2.4}$ (liver: 98 ± 6 s, spleen: 124 ± 19 s), HEM$_{1.5}$ (liver: 108 ± 15 s, spleen: 133 ± 27 s), PEM$_{1.5}$ (liver: 104 ± 9 s, spleen: 148 ± 24 s), indicating that the materials with functional components could efficiently improve control hemorrhage capability (Fig. 4a). The hemostasis efficiency in the in vivo rat tail amputation model showed a similar improvement to that of the rat liver and spleen hemostasis model for NEMs (e.g., from 428 ± 42 s (PVPEM) to 190 ± 51 s (KEM$_{1.5}$)). It is worth noting that KEM$_{1.5}$ showed the best hemostasis capability among all the NEMs, which reaches the 'standard of care' level for commercial products (ClG and CoG) in the three in vivo bleeding models (Fig. 4a), varied from those reported nanoclay-containing composites[13,14,16,19], inorganic nanoparticles[44] and other hemostatic materials[5,6,45,46] that involve poor mechanical strength and inflammation and thrombosis risk due to fine particles or the particles exfoliation from incompact structure (Supplementary Table 2).

The hemostasis applicability was investigated by observing untreated wounds, treated wounds and wounds after removal of the materials (Fig. 4b and Supplementary Fig. 12). There were no residues from the NEMs on the wounds (including a clean wound surface after treated with PVPEM, Fig. 4b), in contrast to numerous residues from the hemostatic powders (Supplementary Fig. 12), indicating the practicality and pasting ability of NEMs. And the wound stopped bleeding (liver: 144 ± 22 s, spleen: 227 ± 30 s) treated with PVPEM and was mildly exudative bleeding after 5 m again that might due to no active hemostatic components (Fig. 4b). Easy use and removal of NEMs play an additional significant role in reducing the risks of wound infection or hemolysis, benefitting from the integrated structures of NEMs. In contrast, the particle residues were difficult to clean and may cause hemolysis or other side effects, which is another major disadvantage of directly using nanoclay for hemostasis (Supplementary Fig. 12 and Supplementary Discussion). The commercial products (ClG and CoG) also exhibited high active component leakage due to poor adhesion, which would necessitate scrupulous debridement[6].

Blood loss, which is a key index for hemostasis capability, was investigated by assessing the dressing weight after hemostasis[19]. In the liver model (Supplementary Fig. 13), the KEM$_{1.5}$ dressing weight after hemostasis was 33 ± 6 mg (initial sample weight: 20 mg), which was lighter than that of the commercial products (ClG: 40 ± 10 mg, CoG: 46 ± 11 mg). The dressing weights in the spleen model showed a similar result to those in the liver model (lightest: KEM$_{1.5}$), and the KEM$_{1.5}$ treatment showed the lowest abdominal cavity lavage fluid relative OD value after hemostasis from liver and spleen wounds, which means less blood loss and a

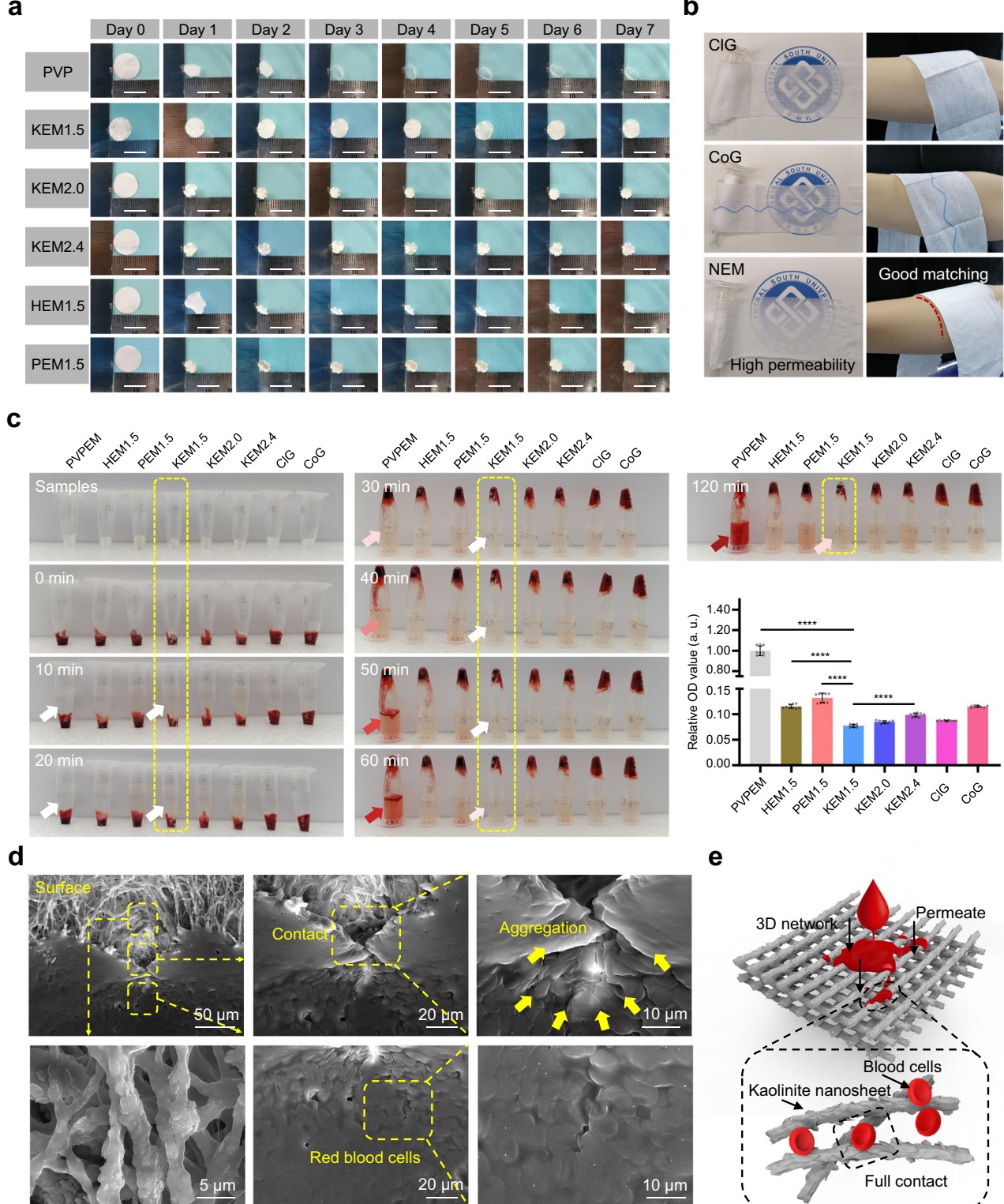

**Fig. 3 Shrinkage and coagulation properties of nanoclay electrospun membranes in vitro. a** Sample 7-day shrinkage tests and results under the condition of constant temperature and humidity for PVPEM, PEM$_{1.5}$, HEM$_{1.5}$, KEM$_{1.5}$, KEM$_{2.0}$ and KEM$_{2.4}$, respectively. Scale bar, 1 cm. **b** Photographs of the KEM$_{1.5}$, CIG and CoG samples for the permeability and application on the skin. **c** Photographs from the in vitro blood-clotting measurement and the corresponding relative OD values of the supernatant absorbance for PVPEM ($n = 9$), HEM1.5 ($n = 9$), PEM1.5 ($n = 9$), KEM$_{1.5}$ ($n = 9$), KEM2.0 ($n = 9$), KEM2.4 ($n = 9$), CIG ($n = 9$), CoG ($n = 9$), respectively. Data were expressed as mean ± s.d.; ****, $p < 0.0001$. Student's $t$ test (two-sided) was used for statistical analysis of between two groups comparison in (**c**). **d** The SEM images of blood cells on the KEM$_{1.5}$ surface. Data were analyzed from at least three independently experiments in (**d**). **e** Illustration of blood components permeation and contact to the KEM1.5. All experiments were replicated at least three times independently with similar results. Source data are provided as a Source Data file.

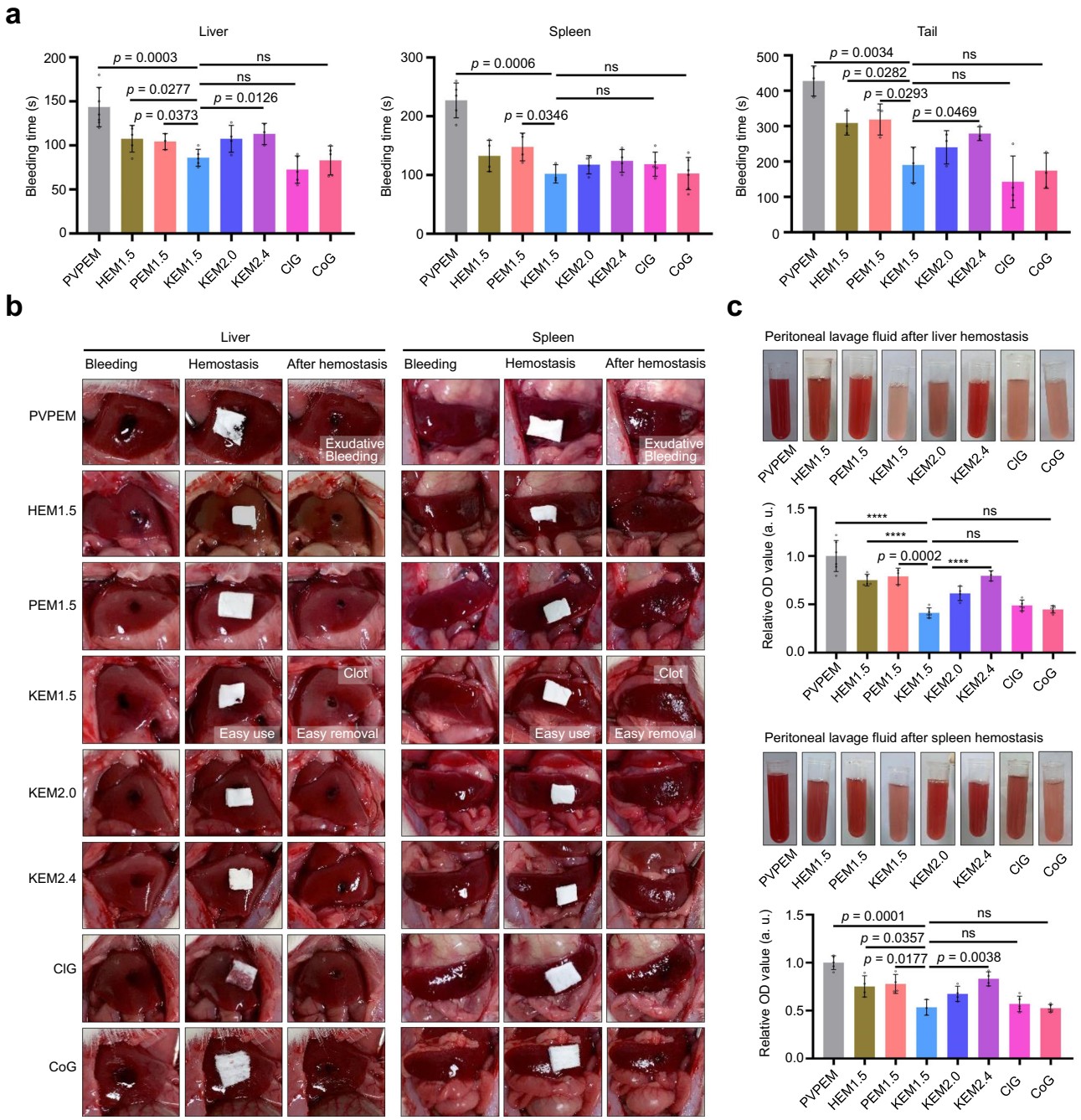

**Fig. 4 In vivo clotting research. a** Bleeding time of wounds treated with NEMs. In rat liver hemostasis model: PVPEM ($n = 7$), HEM$_{1.5}$ ($n = 5$), PEM$_{1.5}$ ($n = 3$), KEM$_{1.5}$ ($n = 5$), KEM$_{2.0}$ ($n = 4$), KEM$_{2.4}$ ($n = 3$), ClG ($n = 5$) and CoG ($n = 5$); in rat spleen hemostasis model: PVPEM ($n = 5$), HEM$_{1.5}$ ($n = 4$), PEM$_{1.5}$ ($n = 4$), KEM$_{1.5}$ ($n = 3$), KEM$_{2.0}$ ($n = 4$), KEM$_{2.4}$ ($n = 4$), ClG ($n = 5$) and CoG ($n = 6$); and in rat-tail amputation hemostasis model: PVPEM ($n = 3$), HEM$_{1.5}$ ($n = 3$), PEM$_{1.5}$ ($n = 3$), KEM$_{1.5}$ ($n = 3$), KEM$_{2.0}$ ($n = 3$), KEM$_{2.4}$ ($n = 3$), ClG ($n = 4$) and CoG ($n = 3$). **b** Photographs for the bleeding wound with untreated, treated and removed PVPEM, HEM$_{1.5}$, PEM$_{1.5}$, KEM$_{1.5}$, KEM$_{2.0}$, KEM$_{2.4}$, ClG and CoG, respectively, in rat liver and spleen hemostasis model. **c** Abdominal cavity lavage fluid photographs and the corresponding relative OD values after NEMs treatment. In rat liver hemostasis model: PVPEM ($n = 6$), HEM$_{1.5}$ ($n = 5$), PEM$_{1.5}$ ($n = 3$), KEM$_{1.5}$ ($n = 5$), KEM$_{2.0}$ ($n = 4$), KEM$_{2.4}$ ($n = 3$), ClG ($n = 5$) and CoG ($n = 5$); and in rat spleen hemostasis model: PVPEM ($n = 5$), HEM$_{1.5}$ ($n = 4$), PEM$_{1.5}$ ($n = 4$), KEM$_{1.5}$ ($n = 3$), KEM$_{2.0}$ ($n = 4$), KEM$_{2.4}$ ($n = 4$), ClG ($n = 5$) and CoG ($n = 5$). Data were expressed as mean ± s.d.; ****, $p < 0.0001$. Student's $t$ test (two-sided) was used for statistical analysis of between two groups comparison in (**a**) and (**c**). All experiments were replicated at least three times independently with similar results. Source data are provided as a Source Data file.

faster procoagulant capability for KEM$_{1.5}$ (Fig. 4c). In addition, the destabilized rat heart rate (HR) after laparotomy and trauma treatments showed a rapid rescued and stabilization treated with NEMs accompanied by stable blood oxygen saturation (SpO$_2$), which provided a statistically significant advantage for retaining a 100% survival rate over 5 weeks of observation after hemostasis

(Supplementary Fig. 14 and Supplementary Discussion), confirming good hemostasis and rescue effects[46].

**Biosafety evaluation.** The biosafety of hemostatic membranes is particularly important. Nanoclay and PVP have excellent physiological inertia and biocompatibility in medical applications. To

further confirm the biosafety of the monolithic materials, PVPEM, HEM$_{1.5}$, PEM$_{1.5}$, KEM$_{1.5}$, KEM$_{2.0}$, and KEM$_{2.4}$ were implanted into subcutaneous tissue on the back of rats (Fig. 5a), and the tests included a control group (filter paper) and a blank group. Images and histological analyses of the injured wounds after the embedding surgery were recorded for 21 days (Fig. 5b, c). After 7 days, the wounds produced subcutaneous nodules in the control group, whereas the wounds repaired normally after implantation with NEMs or when left untreated (blank group). After 21 days, the subcutaneous nodules did not disappear in the control group, while the wounds in the other groups completely recovered with good degradation of NEMs (Fig. 5b, c). The histological data were consistent with the results of direct observation (Fig. 5c). Severe foreign body granuloma and inflammatory reactions were found in the control group embedding sites, while other samples maintained normal levels of histiocytes (Fig. 5c), suggesting that each group of materials (best: KEM$_{1.5}$) has good biological safety and degradation properties[13].

Biotoxicity effect of NEMs to other organs was further investigated, including the heart, lung, stomach, liver, spleen, and kidney (Fig. 5d), which is necessary and may be influenced by the risk of particle leakage and distal thrombosis[5]. KEM$_{1.5}$ subcutaneous embedding treatment for 21 days showed no significant effect on the shape and size of each organ, which were similar to those of normal rats (Fig. 5d). The organs of the rats treated with KEM$_{1.5}$ showed no obvious substantial damage according to histological analysis, indicating no obvious side effects on the circulatory system, respiratory system, digestive system, or metabolic system. Furthermore, coagulation related tests were investigated to explore the effect of NEMs on clotting cascade, concerning the safety of coagulation system. The thrombin time (TT), FIB, prothrombin time (PT), activated partial thromboplastin time (APTT) in all the NEMs were within 15.07–17.33 s, 2.81–3.29 g/L, 8.47–11.37 s and 25.17–35.46 s (Supplementary Fig. 15 and Supplementary Discussion), respectively, all among normal ranges. It means that a general level was maintained on TT, FIB, PT, or APTT treated with NEMs, which indirectly suggested no obvious bleeding or thrombosis risk for local and other surrounding organs, indicating a good safety of coagulation system[6,45]. And it indicated that the hemostatic mechanism of NEMs may not be significantly related to extrinsic (tissue factor) coagulation cascade pathway activation[13,45], which is consistent with the results of Thromboelastogram (Supplementary Fig. 9). Thus, the biological toxicity assay indicated that the NEMs (best: KEM$_{1.5}$) were harmless to the animals and can potentially be used as rapid robust hemostatic bandages for wounds.

In summary, a series of designable nanoclay-organic self-supported membranes were developed, which have the potential to be directly used as acute hemostasis bandages. In KEM, sheet-like kaolinite was uniformly dispersed in and partly exposed on PVP fibers, facilitating the formation of a stabilizing framework and regulating spontaneous shrinkage and degradation of PVP fibers; these properties were superior to those of PVPEM, HEM and PEM. KEM$_{1.5}$, with its enriched hemostatic functional sites, robust framework, and hydrophilic surface, has better comprehensive performances on hemostatic time, hemostatic effect and blood loss compared with those of the other NEMs, ClG and CoG in an in vivo rat tail amputation model and a rat liver and spleen injury model. KEM (best: KEM$_{1.5}$) with a highly hydrophilic framework promotes blood coagulation by rapidly aggregating the blood components, activating the platelets and triggering the intrinsic coagulation pathway. These results indicate that KEM acute hemostasis bandages with high biosafety are excellent and promising candidate materials for compressible hemorrhage control applications. And it is note that the suitability under

various trauma application environment, the long-term availability (e.g., PVP substrate deliquescence) and the biocompatibility in clinically relevant practice require further studies. NEMs with simple synthesis, good flexibility, easy manufacturability and high hemostatic performance demonstrate that this study provides a new strategy for rapid robust hemostatic bandages.

## Methods

**Materials**. Kaolinite was exploited from Maoming, Guangdong Province. Halloysite was exploited from Tongling, Anhui Province. Palygorskite was exploited from Xuyi, Jiangsu Province. Absolute ethanol, PVP (molecular weight: 1,300,000) and Glutaraldehyde (50% in H$_2$O) were purchased from Aladdin-reagent Inc. Calcium chloride (CaCl$_2$) and Phosphate Buffer Saline (PBS, pH = 7.0) was purchased from Hunan Hongjie Chemical Technology Co., Ltd. Antibody CD61-PE/Cyanine7 (CD61-PE/Cy7; clone: VI-PL2; catalog: 336416; lot: B318046) and CD62P-Brilliant Violet 605$^{TM}$ (CD62P-BV605; clone: AK4; catalog: 304920; lot: B317810) were purchased from commercial sources (Biolegend, USA). Quikclot CoG and ClG were purchased from commercial source (Z-Medica, USA).

**Synthesis and shrinkage of nanoclay electrospun membranes**. The theoretical stoichiometry of kaolinite[47], halloysite[48] and palygorskite[29] are Al[Si$_4$O$_{10}$](OH)$_8$, Al$_2$Si$_2$O$_5$(OH)$_4$·nH$_2$O (n = 2 or 4) and Mg$_5$Si$_8$O$_{20}$(OH)$_2$(OH$_2$)$_4$·4H$_2$O (Supplementary Table 1), respectively. It is highlighted that nanoclays physicochemical characteristic cluster analysis and the internal relationship establishment of hemostatic mechanism; representatively, kaolinite was chosen as the main material of hemostat design, palygorskite and halloysite as alternative materials. In this study, a mass ratio was used to prepare the different nanoclay-based electrospun membrane (NEMs) (Supplementary Table 1), which would be applicable in mineral based composite materials. It could be attributed to the complicated composition and water content for practical mineral particles, even under same general stoichiometry.

The electrospun solution was prepared by dispersing nanoclay into 10 mL of absolute ethanol and stirred vigorously for 1 h, followed by the addition of 1 g PVP under magnetic stirring for 10 h at room temperature. Nanoclay membranes were fabricated by an electrospinning device at a total voltage of 15 kV and a steady flow rate of 0.05 μL/min. The various kinds and dosages of nanoclay were controlled to synthesize NEMs (denoted as PVPEM, HEM$_{1.5}$, PEM$_{1.5}$, KEM$_{1.5}$, KEM$_{2.0}$, and KEM$_{2.4}$, respectively), and the detailed ratios are shown in Supplementary Table 1. The shrinkage observation of NEMs was instantly recorded after the electrospinning completement. Each sample with diameters of 1.2 cm was placed in an environment with constant temperature (30 °C) and humidity (40%), which was observed for continuous 7 days shrinkage changes. Then, the area changes were recorded by a smartphone (STK-AL00) at the constant distance and position and were accurately measured and calculated by a software (Nano Measurer 1.2).

**Characterization**. The Diffuse Reflectance Infrared Fourier Transform Spectroscopy were tested by a Shimadzu FTIR 8120 spectrometer at a wavenumber range between 4000 and 600 cm$^{-1}$. The SEM images were examined by a Mira3 LMU SEM (Tescan, Czech Republic) with the voltage of 20 kV. The TEM images and EDX mapping analysis were collected with a Titan G2 60-300 with the voltage of 30 kV. The differential thermal analysis was analyzed using the STA449C (NETZSCH, GER), which ranged from 30 to 1000 °C with 5 °C/min rate of rise. XRD measurements were obtained by Rigaku TTR III at range between 5 and 80°. The absorbance (OD) value was obtained from a microplate reader (EnVision Xcite, PerkinElmer). Zeta potentials of samples were measured with a Zeta-sizer Delsa440sx instrument (Beckman Coulter, Brea, CA, USA). The water contact angle was tested with a contact angle goniometer (JY-82C, Chengde Dingsheng Testing Machine Testing Equipment Co., Ltd.). OriginPro (version 8.5) was used to treated the data for materials properties analysis.

**Blood coagulation in vitro**. Membrane samples (10 mg) were prepared and put in centrifuge tubes; the samples included PVPEM (control), HEM$_{1.5}$, PEM$_{1.5}$, KEM$_{1.5}$, KEM$_{2.0}$, KEM$_{2.4}$ and commercial controls (ClG and CoG). A volume of 100 μL whole blood was dropped onto each sample and then immediately recalcified with 10 μL 0.2 M CaCl$_2$ solution. A volume of 1 mL DI water was gently added into the centrifuge tubes to release unbound blood components (avoiding clot destruction) after 10 min. Then, the centrifuge tubes were turned upside down to investigate the status of the coagulation clot at 30 min. Finally, the absorbance of the supernatant was tested at 540 nm, which is a key criterion for accurately evaluating the unbound blood amount (a higher OD) value indicates a slower coagulation rate). In addition, a volume of 100 μL whole blood was dropped onto 10 mg round membrane samples with a 1 cm diameter, and then blood adsorption, permeation and coagulation were observed directly. 100 μL whole blood was dropped onto 10 mg membrane sample for each group. The samples were added with 10 μL 0.2 M CaCl$_2$ solutions immediately and incubated at 37 °C for 5 min. The prepared samples were chemically fixed using 2.5% glutaraldehyde in PBS (pH = 7.0) at 25 °C for 4 h. Then, the samples were dehydrated with PBS (two times), 25%, 50%, 75%, 80%, 90%, and 100% (two times) ethanol serially (treated 15 min for each

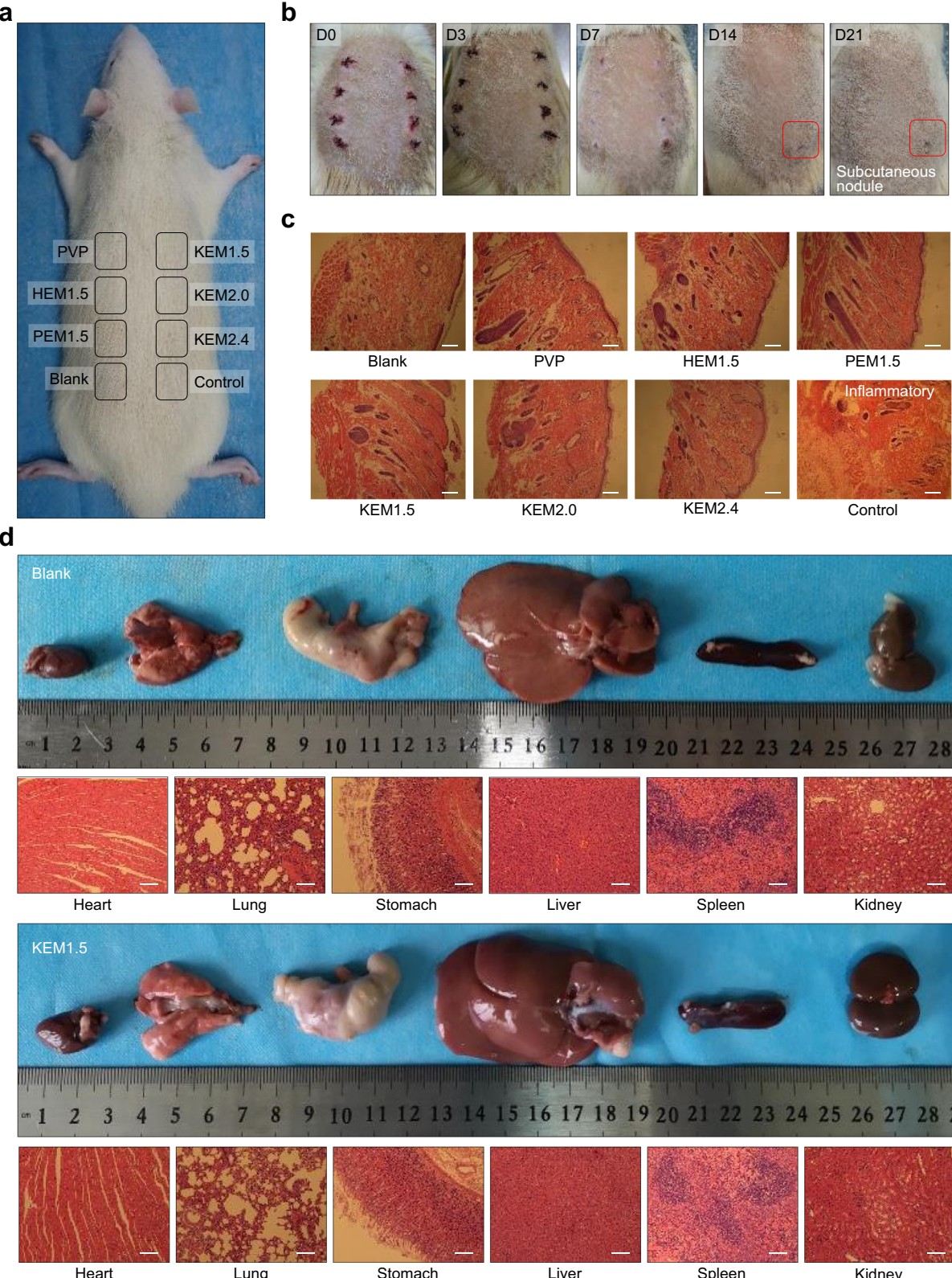

**Fig. 5 Biological toxicity evaluation. a** Photograph for the wound regions in back embedding model. **b** Photographs of the wounds that were embedded the PVPEM, HEM$_{1.5}$, PEM$_{1.5}$, KEM$_{1.5}$, KEM$_{2.0}$, KEM$_{2.4}$, control group and blank group samples immediately following surgery (day 0), and for 3, 7, 14, 21 day. **c** Histological analysis of subcutaneous tissue of the wounds after embedding surgery. Scale bar, 200 μm. **d** Photographs and histological analysis for rat internal organs (heart, lung, stomach, liver, spleen and kidney) after 21 days embedding experiment with KEM$_{1.5}$ and blank group, respectively. Scale bar, 100 μm. Data were analyzed from at least three independently experiments in (**c**) and (**d**). All experiments were replicated at least three times independently with similar results. Source data are provided as a Source Data file.

step). The dried samples were investigated the blood cells contact and aggregation (surface and interior) by SEM.

5 μL fresh whole blood with sodium citrate (9:1) anticoagulant was dropped on 10 mg NEMs membrane sample for each group (incubation for 10 min). 100 μL PBS, 2 μL antibody CD61 (the platelet markers) and 2 μL antibody CD62P (the dominant platelet activation markers)[37,38] were added and bound to the samples then oscillated to fully disperse (incubation for 15 min). The samples were diluted using 200 μL PBS then filtrated the NEMs samples using a filter. The percentage of $CD61^+$ and double $CD61^+/CD62P^+$ cells was analyzed by a flow cytometer (DxP Athena V2-B4-R2, Wuxi Xiatai Biological Technology Co., Ltd.). 100 mg sample was added to 1 mL whole blood with sodium citrate (9:1) anticoagulant for each group (incubation for 10 min). 340 μL prepared whole blood solutions were calcified with 20 μL 0.2 M $CaCl_2$ solution in the test cups. Then, the samples were investigated using a Thromboelastogram analyzer (CFMS LEPU-8800, Lepu Medical Technology (Beijing) Co., Ltd.).

**Visceral hemostasis in vivo assay**. All animal experiments were performed according to the relevant ethical regulations of Central South University, and this study received approval from Central South University Experimental Animal Ethics Committee. All male Sprague Dawley (SD) rats were obtained (220–250 g, 7–8 weeks) from the Hunan SJA Laboratory Animal Co., Ltd (Hunan, China), kept in a dedicated feeding box and randomly divided into different groups. All the animal experiments were performed by specific operators for the accuracy of data reducing artificial error, all procedures of which were performed under aseptic condition. The in vivo hemostatic capability of the NEMs were evaluated by representative rat liver and spleen surface trauma model[6,49] and rat tail amputation model[16].

SD rats were standardly anesthetized with 2% sodium pentobarbital, and then laparotomy was performed to fully expose the liver or spleen in the abdominal cavity. An active bleeding wound of ~0.4*0.4 cm on the liver or spleen surface was made by a scalpel. Filter paper was used to make an indicator bar with a width of 0.2 cm, and a black indicator line was drawn 0.5 cm from the end of the indicator bar. After wound bleeding, the indicator bar was immediately placed on the wound. Then, the blood seeped along the indicator bar. If the blood seep to the black line within 3 s, the speed and initial amount of bleeding were qualified for subsequent hemostatic experiments. After completing the bleeding assessment, the same quantity (20 mg) of PVPEM, $HEM_{1.5}$, $PEM_{1.5}$, $KEM_{1.5}$, $KEM_{2.0}$, $KEM_{2.4}$, ClG, CoG, halloysite particles, palygorskite particles, kaolinite particles or medical gauze (the control group of just compression) was immediately put on the wound surface with slight compression for the hemostasis experiment. The blank control group only underwent an active bleeding wound without any hemostatic treatment. An indicator bar was used intermittently to assess active bleeding on the wound. If there was no obvious penetration of blood along the indicator bar, there was no active bleeding. Then, the total bleeding time was recorded, and the dress weigh was measured instantly to evaluate blood loss.

After the hemostasis experiment, the abdominal cavity was flushed with 10 mL of normal saline. The absorbance value of abdominal cavity lavage fluid was measured with a microplate reader (EnVision Xcite, PerkinElmer) to indirectly evaluate the amount of bleeding. During the experiment, the HR and blood oxygen saturation ($SpO_2$) of rats were monitored by a noninvasive animal blood oxygen instrument (DB15, RocSea). After the experiment was completed, the materials and blood clots were removed, and the abdominal cavity was closed. The 5-week survival rate of rats in each group was observed.

**Tail hemostasis in vivo**. All SD rats were evenly and randomly divided into different groups. Then, all SD rats were anesthetized with sodium pentobarbital, and the tail was cut off 5 cm from the end. The tail was placed vertically. Blood dripping was observed to ensure a constant speed after cutting, and then subsequent hemostatic experiments were performed. After the bleeding assessment was completed, the tail with active bleeding was immediately placed in a 15 mL centrifuge tube containing the same quantity (100 mg) of PVPEM, $HEM_{1.5}$, $PEM_{1.5}$, $KEM_{1.5}$, $KEM_{2.0}$, $KEM_{2.4}$, ClG or CoG to ensure that the wound surface was in contact with the materials with slight compression. The tail was raised intermittently to observe wound bleeding. If the blood volume in the centrifuge tube did not increase significantly, there was no active bleeding due to the completion of hemostasis. Finally, the total time to bleeding stoppage was recorded.

**Biological safety evaluation**. To evaluate the biocompatibility and degradation performance of the NEMs in vivo, SD rats were anesthetized with sodium pentobarbital, and the hair on the back skin was removed to prepare for surgery. Subcutaneous incisions were formed on the back skin, and materials from each group (20 mg) were embedded under the skin. The blank control group only underwent a subcutaneous incision, which was sutured immediately without embedding any materials. Filter paper (FP) was embedded under the skin to form a positive control group to observe foreign body reactions. The skin reactions of each group were observed on days 0, 3, 7, 14, and 21. After 21 days, the rats were euthanized, and the full-thickness skin tissues of each group were taken for hematoxylin-eosin (HE) staining by a microscopy (Leica, Wetzlar, Germany) to observe the skin pathological changes (especially foreign body reactions and

inflammatory reactions) and to evaluate the degradation performance of the materials. In addition, $KEM_{1.5}$ was embedded under the skin alone, and the blank control group was used as a negative control. After 21 days, the rats were euthanized. The heart, lung, stomach, liver, spleen, and kidney were removed to observe the shape and size. Then, the pathological changes in these organs were observed by HE staining to assess whether $KEM_{1.5}$ caused substantial damage to various organs after degradation and absorption.

Adhering to the relevant guidelines and ethical regulations of Central South University, clotting cascade assessment for the NEMs was performed by collecting blood donated from healthy volunteers by a professional doctor at Third Xiangya Hospital of Central South University. 500 μL whole blood was treated with 50 mg material for each group and then analyzed at the Department of Hematology in Third Xiangya Hospital of Central South University for assessment of coagulation function[13]. Detection indicators included APTT, PT, TT, international normalized ratio, fibrinogen concentration (FIB) and prothrombin time activity (PTA).

**Statistics and reproducibility**. The data in this study are presented as mean ± standard deviation (s.d.) and represent a minimum of three independent experiments with at least three technical replicates unless otherwise stated. GraphPad Prism (version 8.0) was used for the statistical analysis with Student's $t$ test (two-sided) between two groups comparison (****, $p < 0.0001$). Values of $p < 0.05$ were considered significant. All experiments in this study were replicated at least three times independently, and all replications showed reproducibly similar data.

**Reporting summary**. Further information on research design is available in the Nature Research Reporting Summary linked to this article.

## Data availability
All data are available within the Article and Supplementary Files, or available from the corresponding authors upon reasonable request. Source data are provided with this paper.

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

# ARTICLE

17. Gogoi, R. K. & Raidongia, K. Strategic shuffling of clay layers to imbue them with responsiveness. *Adv. Mater.* **29**, 1701164 (2017).

18. Li, C., Mu, C., Lin, W. & Ngai, T. Gelatin effects on the physicochemical and hemocompatible properties of gelatin/PAAm/laponite nanocomposite hydrogels. *ACS Appl. Mater. Interfac.* **7**, 18732–18741 (2015).

19. Liang, Y. et al. Graphene-kaolin composite sponge for rapid and riskless hemostasis. *Colloid Surf. B* **169**, 168–175 (2018).

20. Liao, Y., Loh, C.-H., Tian, M., Wang, R. & Fane, A. G. Progress in electrospun polymeric nanofibrous membranes for water treatment: fabrication, modification and applications. *Prog. Polym. Sci.* **77**, 69–94 (2018).

21. Lvov, Y. M., Shchukin, D. G., Mohwald, H. & Price, R. R. Halloysite clay nanotubes for controlled release of protective agents. *ACS Nano* **2**, 814–820 (2008).

22. Isayama, M. & Kunitake, T. Self-supporting films of clay minerals and metal oxides: molecular ceramics. *Adv. Mater.* **6**, 77–78 (1994).

23. Grandgeorge, P. et al. Capillarity-induced folds fuel extreme shape changes in thin wicked membranes. *Science* **360**, 296–299 (2018).

24. Liu, W., Song, M. S., Kong, B. & Cui, Y. Flexible and stretchable energy storage: recent advances and future perspectives. *Adv. Mater.* **29**, 1603436 (2017).

25. Rogers, J. A., Takao, S. & Yonggang, H. Materials and mechanics for stretchable electronics. *Science* **327**, 1603–1607 (2010).

26. Cheng, H. et al. Electrokinetic energy conversion in self-assembled 2D nanofluidic channels with janus nanobuilding blocks. *Adv. Mater.* **29**, 1700177 (2017).

27. Hirsemann, D. et al. Large-scale, low-cost fabrication of janus-type emulsifiers by selective decoration of natural kaolinite platelets. *Angew. Chem. Int. Ed.* **51**, 1348–1352 (2012).

28. Zhang, Y., Tang, A., Yang, H. & Ouyang, J. Applications and interfaces of halloysite nanocomposites. *Appl. Clay Sci.* **119**, 8–17 (2016).

29. Ruiz-Hitzky, E. et al. Fibrous clays based bionanocomposites. *Prog. Polym. Sci.* **38**, 1392–1414 (2013).

30. Hu, P. & Yang, H. Insight into the physicochemical aspects of kaolins with different morphologies. *Appl. Clay Sci.* **74**, 58–65 (2013).

31. Ouyang, J. et al. $CO_2$ capturing performances of millimeter scale beads made by tetraethylenepentamine loaded ultra-fine palygorskite powders from jet pulverization. *Chem. Eng. J.* **341**, 432–440 (2018).

32. Szczepanik, B. et al. The effect of chemical modification on the physico-chemical characteristics of halloysite: FTIR, XRF, and XRD studies. *J. Mol. Struct.* **1084**, 16–22 (2015).

33. Shahhosseininia, M., Bazgir, S. & Joupari, M. D. Fabrication and investigation of silica nanofibers via electrospinning. *Mater. Sci. Eng. C. Mater. Biol. Appl* **91**, 502–511 (2018).

34. Suganya, S., Ram, T. S., Lakshmi, B. S. & Giridev, V. R. Herbal drug incorporated antibacterial nanofibrous mat fabricated by electrospinning: an excellent matrix for wound dressings. *J. Appl. Polym. Sci.* **121**, 2893–2899 (2011).

35. Jin, Y. et al. Buckled Au@PVP nanofiber networks for highly transparent and stretchable conductors. *Adv. Electron. Mater.* **2**, 1500302 (2016).

36. Xue, J. et al. Electrospun microfiber membranes embedded with drug-loaded clay nanotubes for sustained antimicrobial protection. *ACS Nano* **9**, 1600–1612 (2015).

37. Blair, T. A. & Frelinger, A. L. III Platelet surface marker analysis by mass cytometry. *Platelets* **31**, 633–640 (2020).

38. Dave, R. G. et al. Diagnostic utility of flow cytometry based coated-platelets assay as a biomarker to predict thrombotic or hemorrhagic phenotype in acute stroke. *Cytom. Part B-Clin. Cy.* **22026**, 1552–4957 (2021).

39. Velzen, J. F. V., Gorkom, B. A. P. L., Pop, G. A. M. & Heerde, W. L. V. Multicolor flow cytometry for evaluation of platelet surface antigens and activation markers. *Thromb. Res.* **130**, 92–98 (2012).

40. Ortiz-Rivero, S. et al. C3G, through its GEF activity, induces megakaryocytic differentiation and proplatelet formation. *Cell Commun. Signal* **16**, 101 (2018).

41. Differding, J. A. et al. Trauma induces a hypercoagulable state that is resistant to hypothermia as measured by thrombelastogram. *Am. J. Surg.* **201**, 587–591 (2011).

42. Oliver, W. C. *Anticoagulation and Coagulation Management for ECMO* (Sage, 2009).

43. Spiezia, L. et al. Whole blood coagulation assessment using rotation thrombelastogram thromboelastometry in patients with acute deep vein thrombosis. *Blood Coagul. Fibrin.* **19**, 355–360 (2008).

44. Chen, Z. et al. Blood clot initiation by mesoporous silica nanoparticles: dependence on pore size or particle size. *J. Mater. Chem. B* **1**, 1–10 (2013).

45. Wang, X. et al. Exploration of blood coagulation of N-Alkyl chitosan nanofiber membrane in vitro. *Biomacromolecules* **19**, 731–739 (2018).

46. Hsu, B. B. et al. Clotting mimicry from robust hemostatic bandages based on self-assembling peptides. *ACS Nano* **9**, 9394–9406 (2015).

47. Kuroda, Y., Ito, K., Itabashi, K. & Kuroda, K. One-step exfoliation of kaolinites and their transformation into nanoscrolls. *Langmuir* **27**, 2028–2035 (2011).

48. Lvov, Y., Wang, W., Zhang, L. & Fakhrullin, R. Halloysite clay nanotubes for loading and sustained release of functional compounds. *Adv. Funct. Mater.* **28**, 1227–1250 (2016).

49. Pusateri, A. E. et al. Effect of a chitosan-based hemostatic dressing on blood loss and survival in a model of severe venous hemorrhage and hepatic injury in swine. *J. Trauma* **54**, 177–182 (2003).

## Acknowledgements

This work was supported by the NSFC (21878341 and 51804343), the Strategic Priority Research Program of Central South University (ZLXD2017005), the Natural Science Foundation of Hunan Province (2018JJ3670), the Wisdom Accumulation and Talent Cultivation Project of the Third Xiangya Hospital of Central South University (YX202007), the Foundation of Key Laboratory for Palygorskite Science and Applied Technology of Jiangsu Province (HPK201703).

## Author contributions

Y.Z., Q.Z., A.T. and H.Y. conceived the idea and supervised the work. Y.Z., L.L., Y.C., Z.H., and Q.L. designed the experiments and wrote the paper. Y.C., Z.H. and J.J. performed all preparations and characterizations of materials. Q.Z., J.J. and L.L. performed animal experiment and bioinformatics analysis. Y.Z., A.T., H.Y. and Q.L. contributed to analysis of the results. All authors contributed to interpretation of the results, and gave final approval for publication.

## Competing interests

The authors declare no competing interest.
