## [Peer Review File · Nature Communications]

Reviewers' comments:

Reviewer #1 (Remarks to the Author):

The design of new hemostatic biomaterials is of paramount importance for the management and stoppage of acute bleeding. Due to their unique properties, nanoclays have been widely used as an antihemorrhagic agent and for the design of composite hemostatic materials to facilitate blood clotting. In this study, the authors report on the fabrication of nanoclay/polyvinylpyrrolidone (PVP) electrospun membranes for the design a novel organic hemostatic bandage with high hemostasis efficiency. The reported kaolinite-based electrospun bandages showed some potential for stopping acute bleeding in rats. The bandages exhibited a higher hemostatic activity with a lower toxicity to surrounding healthy organs when compared to commercially available products or raw clay powder. However, several experiments lack key controls, no statistical analysis was performed, and the numbers of technical replicates for each experiment are not provided.

Major comments:

- Introduction. The authors should describe in-depth the major differences between their approach and what has been previously reported in the field. For instance, clay-based electrospun fibers have already been investigated and extensively used for several biomedical applications, without any shrinkage reported (main limitation highlighted in the manuscript). Furthermore, several biomaterials using nanoclay are under preclinical investigations, including nanoclay-containing hydrogels. Without these comparisons, it is challenging to assess the novelty of their work.
- Membrane composition. The authors should provide more information on the rationale behind their membrane composition. For instance, unlike kaolinite, only one concentration of Palygorskite and Halloysite was investigated. Also, the authors should better explain why a mass ratio was used to prepare the different nanoclay-based electrospun membrane when a molar ratio would be more scientifically relevant. Substantial molecular weight differences exist between the 3 nanoclay investigated, which could explain the results obtained in this work.
- Hydrophilicity and nanoclay particle dispersion within fibers. At several occasions, the authors commented on the hydrophilicity and dispersion of their particles in electrospun PVP fibers as a function of composition or concentration. However, only shrinking and morphologic observations are described. It is recommended to perform more specific and quantitative assays to validate their hypothesis.
- Plasma absorption and coagulation. First, the technique used to monitor blood coagulation is not clearly described. For some conditions, blood coagulation was monitored by evaluating the volume of blood passing through the membranes and quantifying the membrane to blood volume ratio (Figure 3C). However, based on the pictures shown in the manuscript, it is difficult to tell whether blood coagulated or just dried. In addition, controls including commercially available products (e.g., gauze) or other hemostatic materials should be included for comparison.
- In vivo hemostatic performance. It is unclear how the bleeding time was monitored. As a result, it is difficult for the reader to evaluate the data depicted in Figure 4. There are a number of unanswered questions such as (i) were the devices removed at given time points to quantify bleeding time? (ii) was the initial bleeding quantified to normalize the data? Also, the authors should elaborate more on the generated results. For instance, why blank and PVPEM membranes have the same hemostatic time in spleens (Figure 4C), but led to a bleeding and clean wound respectively? Furthermore, the in vitro data seem to be inconsistent with their in vivo data. These further questions the accuracy of their coagulation data, especially for their HEM membranes.

Minor comments:

- In the Material section, the authors should provide more details regarding their experiments. For

instance, the parameters used for SEM and DTA characterizations are missing and how the wound condition was scored is not provided.

- For the histological (Figure 5C) and rat viscera (Figure 5D) analyses, more information should be provided with respect to treatment duration prior to tissue/organ sampling.
- The authors should mention how many replicates and technical replicates have been used in each experiment. In addition, a statistical analysis of their data should be performed, especially in Figures 3, 4, and 5.
- The manuscript needs some editing. There are several misused/mispelled words (e.g. cly instead of clay), and some lengthy sentences need to be revised.

Reviewer #2 (Remarks to the Author):

The authors present the manufacturing of nano clay-impregnated electrospun PVP systems as innovative hemostatic dressings. The preparation/manufacture of the technology and its physico-chemical characterization is very well-described. However, in the context of hemostatic evaluation of the technology, the manuscript is lacking some significant information, experiments and analysis, as follows:

1. The title is very confusing, with the word 'hemostatic' repeated. The 'Rapid Hemostatic' part can be omitted.
2. The authors set up the context of their research in the framework of 'acute hemorrhage is one of the greatest causes of death'. While this is true, this context is actually more for 'non-compressible acute hemorrhage' (internal organ damage, diffuse injury, poly trauma etc.) and not that much for external accessible (often compressible) hemorrhaging injuries. This context should be distinguished in the Intro, especially since the technology that the authors describe is not for non-compressible hemorrhage.
3. Kaolin/Zeolite-embedded hemostat dressings are already available (e.g. Combat Gauze), that have significant hemostatic efficacy. In this framework, it is unclear what advantage or performance enhancement the described electrospun nanoclay-PVP systems are providing.
4. In the in vivo evaluation of hemostatic efficacy, while the models are well-described (and well-known) models of bleeding, the comparison groups do not include any commercially available well-established 'standard of care' (e.g. QuikClot or Combat Gauze), and also does not include a control group of just 'compression'. These are necessary to demonstrate/establish what hemostatic advantage do the described nanoclay-PVP materials provide beyond 'standard of care' commercial products that are already out there. If such 'standard of care' are not available in the country/location of the reported research, then it should be clearly stated.
5. The only 'quantitative data' provided is 'hemostats time' and there is no statistical analysis/difference described between any of the groups compared. For hemostatic analysis, one should additionally analyze 'blood loss' (e.g. by weighing the dressing) and also analyze the 'clotting mechanisms' (e.g. cellular and molecular components of the clotting process.. like platelet activation, thrombin generation, fibrin content, hemolytic aspects etc.).

6. The impact of a hemostatic technology is not only in 'reducing blood loss' but resultantly stabilizing vitals and improving survival time. It is unclear if any such measurement and analysis was done in the in vivo model. Ideally, if it is a heavy bleeding model from liver and/or spleen, one should observe rapid destabilization of mean arterial pressure, SpO2 etc., and with hemostat stabilization these aspects would be 'rescued'. One should also observe and record 'lethality' (survival), if the model is a lethal model. None of this was reported in the manuscript and so it is unclear what the hemostatic impact actually is.

7. Since this is a localized (and not systemic) application, the rationale behind testing multi-organ toxicity is confusing. Testing locally the inflammatory aspects, as well as the potential of exothermic reactions (if any) from the clay on the tissue should be the focus of the tests.

The responses to the reviewer's comments are shown as follows:

Response to Reviewer 1

The design of new hemostatic biomaterials is of paramount importance for the management and stoppage of acute bleeding. Due to their unique properties, nanoclays have been widely used as an antihemorrhagic agent and for the design of composite hemostatic materials to facilitate blood clotting. In this study, the authors report on the fabrication of nanoclay/polyvinylpyrrolidone (PVP) electrospun membranes for the design a novel organic hemostatic bandage with high hemostasis efficiency. The reported kaolinite- based electrospun bandages showed some potential for stopping acute bleeding in rats. The bandages exhibited a higher hemostatic activity with a lower toxicity to surrounding healthy organs when compared to commercially available products or raw clay powder. However, several experiments lack key controls, no statistical analysis was performed, and the numbers of technical replicates for each experiment are not provided.

1. Introduction. The authors should describe in-depth the major differences between their approach and what has been previously reported in the field. For instance, clay-

based electrospun fibers have already been investigated and extensively used for several biomedical applications, without any shrinkage reported (main limitation highlighted in the manuscript). Furthermore, several biomaterials using nanoclay are under preclinical investigations, including nanoclay-containing hydrogels. Without these comparisons, it is challenging to assess the novelty of their work.

Reply: It was emphasized that kaolinite based electrospun membranes (KEM) with the integrated robust structure of well-dispersed and partially exposed kaolinite particles on electrospun fiber surface in this study, providing more functional hemostatic sites, easier contact with blood components and faster coagulation capability. Polyvinylpyrrolidone electrospun membrane (PVPEM) shrunk sharply (> 75 % shrinkages ratio), due to solvent evaporation and the degradation of PVP, which was inherent spontaneous shrinkage^{23,34,35}. Importantly, nanoclay based hemostatic membrane (NEM) frameworks could resist the shrinkage effectively due to the membrane framework supported by nanoclay fillers (best for KEM_{1.5}, < 30 % shrinkages ratio) (Figure 3a and Figure S8).

NEMs displayed a better hemostatic performance (best for KEM_{1.5}: 86 ± 10 s in rat liver model and 102 ± 16 s in rat spleen model) compared with previously reported biomaterials in Table S2 and Figure 4a. For example, nanoclay-containing hydrogels (*e.g.* gelatin/PAAm/laponite hydrogels¹⁸) and sponges (*e.g.* graphene-kaolinite or graphene-montmorillonite composite sponge^{14,19}) show effective hemostasis however restriction on poor mechanical strength due to the inherent interconnected macroporous structures. Inorganic nanoparticles (like CNTs) or commercial products QuikClot (zeolite) and Combat Gauze (kaolinite) are limited by inflammation and thrombosis risk due to fine particles or the particles exfoliation from incompact structure.

Representative nanoclay based and other hemostasis materials were summarized in Table S2, and the introduction and discussion section have been revised and supplemented as follow.

Page 3 line 22: A few types of biomaterials using nanoclays have been developed as effective hemostats, including hydrogels (e.g., gelatin/polyacrylamide (PAAm)/laponite hydrogels¹⁸) and sponges (e.g., graphene-kaolinite or graphene-montmorillonite composite sponge^{14,19}), which are strict and complicated treatments with high costs as medical hemostats. Hydrogels and sponges show good shape memory but poor mechanical strength due to their inherent interconnected macroporous structures, and polymer binders fill and block the clay pores and layer gaps, causing low clay utilization as active components^{6,16}. Electrostatic spinning is one of the most commonly used membrane synthesis methods; this method can be incorporated with nanoclays and enables the easy preparation of light, fluffy, and soft membrane materials²⁰⁻²². Pure polymer electrospun membranes will sharply spontaneously shrink within a short time after preparation²³, similar to the contraction of cell membranes, which may have potential applications in emerging technologies (flexible electrodes²⁴ or stretchable materials^{25,26}); however, reducing the effective area of the membrane will increase costs and decrease hemostasis capability.

Page 8 line 13: At constant temperature and humidity, the NEM frameworks could effectively resist shrinkage (best for KEM_{1.5}, < 30% shrinkage ratio), while PVPEM shrank sharply (> 75% shrinkage ratio) after the 7-day shrinkage test (Figure S8 and Supplementary Discussion), which was similar to the contraction of cell membranes for preservation²³. The inherent spontaneous shrinkage might be related to solvent evaporation and the degradation for PVPEM^{34,35}. Importantly, the NEMs were well retained without degradation, while PVPEM was almost completely degraded after 2 days of exposure to air (Figure 3a). The good resistance to shrinkage and degradation benefited from synergy between the PVP fibers and kaolinite: The kaolinite nanosheets actively stabilize the framework and regulate the spontaneous shrinkage and degradation of PVP fibers through intermolecular interactions with the PVP fibers²³, and the PVP fibers supply a robust framework for the kaolinite particles³⁴⁻³⁶.

Table S2 Representative examples of various materials recently reported for rapid hemostasis applications

Hemostasis materials and the main components		Benefits	Remarks (specific characteristics, applications and limitations)	Reference
Commercially available products	QuikClot (zeolite) and Combat Gauze (kaolinite)	High porosity, large specific surface area and good biocompatibility facilitating to the blood coagulation through absorbing plasma components as well as easily for accessibility and applicability	Fast hemostatic performance but the exothermicity (for zeolite) and the poor adhesion active component and substrate existing the risk of residual and difficulty for removal from the wounds as well as restriction on internal tissue hemorrhage application	--
Other uncommercialized materials	N-Alkyl chitosan nanofiber membrane	Activation of coagulation factors and platelets for blood coagulation in vitro	Good effects for minor bleeding but the poor mechanical properties and adhesivity unsuitable for severe bleeding	1
	Self-assembling peptides (RADA16-I)	Physical entrapment of blood components in a network of peptide	Excellent performance for hemostasis but the complex fabrication process and expensive cost restriction on practical application	2
	CNT-reinforced glycidyl methacrylate (GMA) functionalized quaternized chitosan (QCSG/CNT)	Injectable and conductive shape memory cryogels presents excellent hemostatic performance in rabbit liver defect lethal noncompressible hemorrhage model (Time: ~3.9 min)	Suitable for irregularly shaped hemorrhage but the residual CNT existing the risk of thrombus and difficulty for removal from wound after bleeding-stop	3
	Mesoporous silica nanoparticles (MSN)	Promoting the blood proteins to contact the huge interior surfaces of MSN and then initiate the quick blood clot	Accessibility and diffusion of clotting-promoting proteins for hemostasis but the fine MSN particles existing the risk of thrombus and histogenic immunity	4
	Mesoporous zeolite-cotton hybrid hemostat (mCHA-C)	Mesoporous zeolite onto the surface of cotton fiber by on-site template-free growth with higher procoagulant activity and minimized loss of active components in rabbit lethal femoral artery injury model (Time: 159 ± 12 s)	Great effects for severe hemorrhage but the complicated fabrication process restricted practical application and existing risk of secondary damage on the wound from exothermicity	5
	Gelatin/PAAm/Laponite Nanocomposite Hydrogels	Hierarchical structures, high transparencies, good homogeneity and ultrastrong tensibilities	Less toxic effects but restriction on severe hemolysis, poor mechanical strength and long clotting time	6
	Graphene-montmorillonite or graphene-kaolin composite sponge (GMCS)	Quickly absorbing plasma to increase the concentration of hemocytes and platelets in rabbit artery injury test (Time: ~85 s)	Good biocompatibility but the exfoliated and residual MMT or kaolin particles difficulty for removal and degradation from bleeding tissue site	7,8
This work	Integrated Nanoclay Electrospun Membranes (best: KEM _{1.5})	Accelerate the coagulation of blood through rapidly absorbing plasma and aggregating blood components with a highly hydrophilic framework, and FXII could be activated by kaolinite on the integrated fiber surface to trigger the intrinsic coagulation	The NEMs with the integrated structure and robust framework have excellent hemostatic capability with low biotoxicity	--

Figure 3a Sample 7-day shrinkage tests and results under the condition of constant temperature and humidity for PVPEM, PEM_{1.5}, HEM_{1.5}, KEM_{1.5}, KEM_{2.0} and KEM_{2.4}, respectively.

Figure 4a Bleeding time of wounds treated with PVPEM (n = 7 or 5 or 3), HEM_{1.5} (n = 5 or 4 or 3), PEM_{1.5} (n = 3 or 4), KEM_{1.5} (n = 5 or 3), KEM_{2.0} (n = 4 or 3), KEM_{2.4} (n = 3 or 4), ClG (n = 5 or 4) and CoG (n = 5 or 6 or 3) in rat liver, spleen, rat-tail amputation hemostasis model, respectively. All hemostasis experiments were replicated at least three times (n ≥ 3) independently with similar results.

14. Li, G., Quan, K., Liang, Y., Li, T., Yuan, Q., Tao, L., Xie, Q. & Wang, X. Graphene-Montmorillonite Composite Sponge for Safe and Effective Hemostasis. *ACS Appl. Mater. Inter.* **2016**, 8 (51), 35071-35080.

18. Li, C., Mu, C., Lin, W. & Ngai, T. Gelatin Effects on the Physicochemical and Hemocompatible Properties of Gelatin/PAAm/Laponite Nanocomposite Hydrogels. *ACS Appl. Mater. Interfaces* **2015**, 7, 18732-18741.

19. Liang, Y., Xua, C., Lia, G., Liub, T., Liangb, J. F. & Wang X. Graphene-kaolin composite sponge for rapid and riskless hemostasis. *Colloid Surface B* **2018**, *169*, 168-175.
23. Grandgeorge, P., Krins, N., Hourlier-Fargette, A., Laberty-Robert, C., Neukirch, S. & Antkowiak, A. Capillarity-Induced Folds Fuel Extreme Shape Changes in Thin Wicked Membranes. *Science* **2018**, *360* (6386), 296-299.
34. Suganya, S., Ram, T. S., Lakshmi, B. S. & Giridev, V. R. Herbal drug incorporated antibacterial nanofibrous mat fabricated by electrospinning: an excellent matrix for wound dressings. *J. Appl. Polym. Sci* **2011**, *121*, 2893-2899.
35. Jin, Y., Hwang, S., Ha, H., Park, H., Kang, S.-W., Hyun, S., Jeon, S. & Jeong, S.-H. Buckled Au@PVP nanofiber networks for highly transparent and stretchable conductors. *Adv. Electron. Mater.* **2016**, *2*, 1500302.

2. Membrane composition. The authors should provide more information on the rationale behind their membrane composition. For instance, unlike kaolinite, only one concentration of Palygorskite and Halloysite was investigated. Also, the authors should better explain why a mass ratio was used to prepare the different nanoclay-based electrospun membrane when a molar ratio would be more scientifically relevant. Substantial molecular weight differences exist between the 3 nanoclay investigated, which could explain the results obtained in this work.

Reply: Clays (most: kaolinite) were widely accepted as effective hemostatic components and applied on commercial hemostats, such as Z-Medica products: Clotting Gauze (ClG) and Combat Gauze (CoG). The theoretical stoichiometry of kaolinite⁴⁰, halloysite⁴¹ and palygorskite²⁹ are $\text{Al}[\text{Si}_4\text{O}_{10}](\text{OH})_8$, $\text{Al}_2\text{Si}_2\text{O}_5(\text{OH})_4 \cdot n\text{H}_2\text{O}$ ($n=2$ or 4) and $\text{Mg}_5\text{Si}_8\text{O}_{20}(\text{OH})_2(\text{OH}_2)_4 \cdot 4\text{H}_2\text{O}$, respectively. It is highlighted that nanoclays physicochemical characteristic cluster analysis and the internal relationship establishment of hemostatic mechanism; representatively, kaolinite was chosen as the main material of hemostat design, palygorskite and halloysite as alternative materials.

Mass ratio would be more applicable than molar ratio in mineral based composite materials with previously literatures, such as Jie F.⁶ (*Nat. Commum* **2019**, *9*, 09849), Lvov, Y³⁶ (*Acs Nano* **2015**, *9* (2), 1600-1612) and Yang, H.¹³ (*Adv. Funct. Mater.* **2018**, *28* (10), 1704452), which might be attributed to the complicated composition and water content even under same general stoichiometry for practical mineral particles. Importantly, the enhanced hemostatic capability was not only from substantial molecular weight differences, but more benefited from the enriched hemostatic functional sites, robust fluffy framework and hydrophilic surface for KEM. The detailed descriptions of membrane composition have been added in introduction and experiments part as follow.

Page 3, line 10: Zeolites and clays, the main components of Z-Medica products (Clotting Sponges, Clotting Gauze (ClG) and Combat Gauze (CoG)), are widely accepted as effective hemostats^{6,13}. Clays can accelerate the stoppage of bleeding by cooperating with physical hemostatic functions, which facilitate plasma absorption to increase blood cell concentration, and with endogenous hemostasis pathways for negatively charge-stimulated clotting factors¹³⁻¹⁵.

Page 14, line 20: The theoretical stoichiometry of kaolinite⁴⁰, halloysite⁴¹ and palygorskite²⁹ are $Al[Si_4O_{10}](OH)_8$, $Al_2Si_2O_5(OH)_4 \cdot nH_2O$ (n=2 or 4) and $Mg_5Si_8O_{20}(OH)_2(OH_2)_4 \cdot 4H_2O$ (Table S1), respectively. It is highlighted that nanoclays physicochemical characteristic cluster analysis and the internal relationship establishment of hemostatic mechanism; representatively, kaolinite was chosen as the main material of hemostat design, palygorskite and halloysite as alternative materials. In this study, a mass ratio was used to prepare the different nanoclay-based electrospun membrane (NEMs) (Table S1), which would be applicable in mineral based composite materials referencing previously literatures^{6,11,36}. It might be attributed to the complicated composition and water content for practical mineral particles, even under same general stoichiometry.

6. Yu, L., Shang, X., Chen, H., Xiao, L., Zhu, Y. & Fan, Jie. A tightly-bonded and flexible mesoporous zeolite-cotton hybrid hemostat. *Nat. Commum* **2019**, *9*, 09849.
13. Long, M., Zhang, Y., Huang, P., Chang, S., Hu, Y., Yang, Q., Mao, L. & Yang, H. Emerging Nanoclay Composite for Effective Hemostasis. *Adv. Funct. Mater.* **2018**, *28* (10), 1704452.
29. Ruiz-Hitzky, E., Darder, M., Fernandes, F. M., Wicklein, B., Alcântara, A. C. S. & Aranda, P. Fibrous Clays Based Bionanocomposites. *Prog Polym. Sci.* **2013**, *38* (10-11), 1392-1414.
36. Xue, J., Niu, Y., Gong, M., Shi, R., Chen, D., Zhang, L. & Lvov, Y. Electrospun Microfiber Membranes Embedded with Drug-Loaded Clay Nanotubes for Sustained Antimicrobial Protection. *Acs Nano* **2015**, *9* (2), 1600–1612.
40. Kuroda, Y., Ito, K., Itabashi, K. & Kuroda, K. One-Step Exfoliation of Kaolinites and Their Transformation Into Nanoscrolls. *Langmuir* **2011**, *27* (5), 2028-35.
41. Lvov, Y., Wang, W., Zhang, L. & Fakhrullin, R. Halloysite Clay Nanotubes for Loading and Sustained Release of Functional Compounds. *Adv. Funct. Mater.* **2016**, *28*, 1227-1250.

3. Hydrophilicity and nanoclay particle dispersion within fibers. At several occasions, the authors commented on the hydrophilicity and dispersion of their particles in electrospun PVP fibers as a function of composition or concentration. However, only shrinking and morphologic observations are described. It is recommended to perform more specific and quantitative assays to validate their hypothesis.

Reply: The SEM images, TEM and mapping analysis have been supplemented to better represent the dispersion of nanoclays. It is clearly showed that most kaolinite particles were evenly dispersed in the fibers and partly exposed to the fiber surface due to their individual sheet-like structures (Figure 2c (I)); while the tube-like halloysite and rod-like palygorskite were trapped in the radial direction along the fibers. The incorporation and well-dispersed configuration for nanoclay was crucial to hydrophilicity improvement, such as, water contact angle decrease from 100° to 62° with increased kaolinite mass ratio in KEM (Figure S4). More detailed description sentences of

dispersion and hydrophilicity of nanoclays have been added in introduction and discussion section as follows.

Page 7 line 16: Single fibers in the NEMs were investigated by SEM, TEM and EDX element analysis to assess the nanoclay particle dispersion and inner structure characteristics. Most kaolinite particles were totally trapped in the fibers with large and evenly diameters (especially KEM_{1.5}, Figure S5 and Supplementary Discussion), and partly exposed to the fiber surface due to their individual sheet-like structures (Figure 2c (I-III)); this configuration was crucial to blood contact and accelerated coagulation. The polymer layer became thinner with increasing kaolinite particle ratio (Figure 2c (II, III)), resulting in weak adhesion interactions and thus increasing the risk of thrombus from particle exfoliation⁶. The tube-like halloysite and rod-like palygorskite were trapped in the radial direction along the fibers and rarely exposed to the surface (shown clearly in Figure 2c (I-III)), restricting blood contact. Furthermore, the element analysis results (Figure S6 and Supplementary Discussion) match well with the distribution and structure characteristics from SEM and TEM analysis, indicating excellent hemostasis potential for KEM (best: KEM_{1.5}) due to its integrated structure and self-supported 3D framework.

Supporting Information Page 16 line 1: The static water-contact angles show great decrease from PVPEM (100°) to the NEMs (< 71°), indicating the improvement of hydrophilicity due to incorporation of nanoclays (Figure S4). The KEM showed an obvious improvement of hydrophilicity (contact angle from 62° to 31°) with the increase additive mass ratio of kaolinite (from 60 % to 70.6 %), indicating that hydrophilicity was related to the clay mass ratio in NEMs.

Figure 2 (c) (I) The SEM images of single-layer membranes (including the single fiber in the inset), (II) TEM and (III) mapping analysis images for PVPEM, HEM_{1.5}, PEM_{1.5}, KEM_{1.5}, KEM_{2.0}, and KEM_{2.4}, respectively. Scale bars in (I–III) were 10, 0.5, 1 μm , respectively. Scale bars in the insets of (I), 2 μm .

Figure S4. Water contact angles for PVPEM, PEM_{1.5}, HEM_{1.5}, KEM_{1.5}, KEM_{2.0} and KEM_{2.4}.

4. Plasma absorption and coagulation. First, the technique used to monitor blood coagulation is not clearly described. For some conditions, blood coagulation was monitored by evaluating the volume of blood passing through the membranes and quantifying the membrane to blood volume ratio (Figure 3C). However, based on the pictures shown in the manuscript, it is difficult to tell whether blood coagulated or just dried. In addition, controls including commercially available products (e.g., gauze) or other hemostatic materials should be included for comparison.

Reply: The experiment of blood clotting in vitro was supplemented by referencing the related literatures^{5,6} to better evaluate blood adsorption, permeation and coagulation for the NEMs (including commercial controls); the detailed methods have been added in the experiment section.

The results clearly showed that KEMs have better blood permeation and coagulation capability (best for KEM_{1.5}, < 2 min) than that of other samples (Figure 3c and Figure S9), meeting the 'standard of care' level for the commercial controls (CoG and ClG). The KEM could effectively accelerate blood coagulation in vitro with a steady state of clot (best: KEM_{1.5}, almost no hemolysis); while the whole clot with PVPEM was finally hemolytic with relative supernatant absorbance (OD) value of 0.620±0.029 that was more than 12 times that of KEM_{1.5} (min: 0.048±0.002) (Figure 3c). A similar result was clearly showed from direct observation (Figure S9), and the details of plasma absorption and coagulation analysis have been added as follows.

Page 16 line 13 (**methods**): Considering all aspects of blood clotting time measurement in vitro, experiments were designed by referencing the related literatures to evaluate blood adsorption, permeation and coagulation for the NEMs^{5,6}. Membrane samples (10 mg) were prepared and put in centrifuge tubes; the samples included PVPEM (control), HEM_{1.5}, PEM_{1.5}, KEM_{1.5}, KEM_{2.0}, KEM_{2.4} and commercial controls (Clotting Gauze (ClG) and Combat Gauze (CoG)). A volume of 100 µL whole blood was dropped onto each sample and then immediately recalcified with 10 µL 0.2 M CaCl₂ solution. A volume of 1 mL DI water was gently added into the centrifuge tubes to release unbound blood components (avoiding clot destruction) after 10 min. Then, the centrifuge tubes were turned upside down to investigate the status of the coagulation clot at 30 min. Finally, the absorbance of the supernatant was tested at 540 nm, which is a key criterion for accurately evaluating the unbound blood amount (a higher optical density (OD) value indicates a slower coagulation rate). Additionally, a volume of 100 µL whole blood was dropped onto 10 mg round membrane samples with a 1 cm diameter, and then blood adsorption, permeation and coagulation were observed directly. The status of blood cells on NEMs was investigated by SEM¹⁶.

Page 9 line 4: The hemostasis capability of the NEM frameworks was investigated by an in vitro blood coagulation test and assessment of the blood absorption ability⁵ (Figure 3c and Figure S9). The results clearly showed that KEM and the commercial controls (CoG and ClG) could facilitate blood permeation and coagulation within 2~5 min (best for KEM_{1.5}, < 2 min), while blood was difficult to absorb in PVPEM (control), HEM_{1.5}, and PEM_{1.5} within 5 min (Figure S9 and Supplementary Discussion). KEM_{1.5} showed fewer unbound blood components than did the other samples (Figure 3c), which is a quality criterion of blood clotting indicating low hemolysis.

The coagulation capability and status of the clot were investigated by continuous experimental observation and evaluation of the absorbance (OD) of the supernatant^{5,16}. The clot after NEM treatment showed a steady state (best: KEM_{1.5}, almost no hemolysis) after 120 min, while the clot with PVPEM showed gradual hemolysis, and almost the whole clot was finally hemolytic (Figure 3c), indicating poor coagulation capability in vitro without active components. The supernatant relative absorbance (OD) value of the PVPEM sample was considerably higher than those of the other samples (more than 12 times that of KEM_{1.5}), indicating that KEM (best: KEM_{1.5}) could effectively accelerate blood coagulation in vitro.

Supporting Information Page 17 line 4: The photograph showed that the blood could not be absorbed and clot in PVPEM, PEM_{1.5}, and HEM_{1.5} within 5 min; however, the membranes showed a small wrinkle due to formation of the intermolecular hydrogen bonds with PVP by the water molecules of the blood (Figure S9). The blood infiltrated into KEM_{1.5} and KEM_{2.0} (KEM_{2.4}) within one and two minutes and coagulated within two and five minutes, respectively, benefiting from the hydrophilicity and easy contact with the active particles on KEM surface. In blank group, the blood (with the action of EDTA) was clotted after 60 min, indicating that the clotting time is shortened by 12-30-fold in KEM (best: KEM_{1.5}).

Figure 3c Photographs from the in vitro blood-clotting measurement and the corresponding relative OD values of the supernatant absorbance for (I) PVPEM, (II) HEM_{1.5}, (III) PEM_{1.5}, (IV) KEM_{1.5}, (V) KEM_{2.0}, (VI) KEM_{2.4}, (VII) ClG, (VIII) CoG, respectively. (n=9) *****P* < 0.0001 using Student's t-test (two-sided). The error bars represent SD. Source data are provided as a Source Data file.

Figure S9. Photographs for in vitro blood coagulation experiment of PVPEM, HEM_{1.5}, PEM_{1.5}, KEM_{1.5}, KEM_{2.0}, KEM_{2.4} and blank group. Scale units, mm.

5. Zhao, X., Guo, B., Wu, H., Liang, Y. & Ma, P. X. Injectable antibacterial conductive nanocomposite cryogels with rapid shape recovery for noncompressible hemorrhage and wound healing. *Nat. Commun* **2018**, *9*, 2784.

6. Yu, L., Shang, X., Chen, H., Xiao, L., Zhu, Y. & Fan, Jie. A tightly-bonded and flexible mesoporous zeolite-cotton hybrid hemostat. *Nat. Commum* **2019**, *9*, 09849.

5. In vivo hemostatic performance. It is unclear how the bleeding time was monitored. As a result, it is difficult for the reader to evaluate the data depicted in Figure 4. There are a number of unanswered questions such as (i) were the devices removed at given time points to quantify bleeding time? (ii) was the initial bleeding quantified to normalize the data? Also, the authors should elaborate more on the generated results. For instance, why blank and PVPEM membranes have the same hemostatic time in spleens (Figure 4C), but led to a bleeding and clean wound respectively? Furthermore, the in vitro data seem to be inconsistent with their in vivo data. These further questions the accuracy of their coagulation data, especially for their HEM membranes.

Reply: An indicator bar was used intermittently to assess active bleeding on the wound. When there was no penetration of blood, the accurate bleeding time was recorded. The speed and initial amount of bleeding were quantified and controlled in a same level for normalizing the in vivo hemostatic tests, which were assessed by the distance of initial blood seeping to an indicator bar within 3 s. All in vivo hemostatic tests were strictly performed with relevant standards and literatures^{6,16} and the detailed test methods in vivo hemostasis have been added in experiment part as follows.

The in vivo hemostatic experiments were replicated to ensure the data accuracy, the result of which was rearranged in Figure 4 and Figure S11. Without any hemostat or material, the active bleeding would not stop over 5 minutes in a blank group (including both spleen and liver wound); while easy removal of the PVPEM on the wound, it was showed a clean surface without residues (than that of powders hemostat, Figure S11) and exudative bleeding again after 5 minutes due to no active hemostatic component (Figure 4b). The replicated experiments were performed scrupulously and, importantly, exhibited a consistent result that improved in vivo and in vitro hemostatic

capability of NEMs (including HEM_{1.5}). The elaborations of in vivo hemostatic performance have been supplemented in result section as follows.

For **test methods** on Page 17 line 16:

An active bleeding wound of approximately 0.4*0.4 cm on the liver or spleen surface was made by a scalpel. Filter paper was used to make an indicator bar with a width of 0.2 cm, and a black indicator line was drawn 0.5 cm from the end of the indicator bar. After wound bleeding, the indicator bar was immediately placed on the wound. Then, the blood seeped along the indicator bar. If the blood seep to the black line within 3 s, the speed and initial amount of bleeding were qualified for subsequent hemostatic experiments. After completing the bleeding assessment, the same quantity (20 mg) of PVPEM, HEM_{1.5}, PEM_{1.5}, KEM_{1.5}, KEM_{2.0}, KEM_{2.4}, ClG, CoG, halloysite particles, palygorskite particles, kaolinite particles or medical gauze (the control group of just compression) was immediately put on the wound surface with slight compression for the hemostasis experiment. The blank control group only underwent an active bleeding wound without any hemostatic treatment. An indicator bar was used intermittently to assess active bleeding on the wound. If there was no obvious penetration of blood along the indicator bar, there was no active bleeding. Then, the total bleeding time was recorded, and the dress weigh was measured instantly to evaluate blood loss¹⁴.

For **elaborations of in vivo hemostatic performance** Page 10 line 8:

When active bleeding wounds (the initial bleeding was assessed to ensure a same test condition, Figure S10) of the liver and spleen occurred, the bleeding time was more than 300 s without hemostatic material treatment. It showed a long hemostatic time for the control group (common medical gauze: liver, 164 ± 56 s; spleen, 201 ± 52 s, Figure S11) and PVPEM group (liver: 144 ± 22 s, spleen: 227 ± 30 s). The bleeding times showed clear decreases with NEM treatment: KEM_{1.5} (liver: 86 ± 10 s, spleen: 102 ± 16 s), KEM_{2.0} (liver: 108 ± 15 s, spleen: 118 ± 16 s) and KEM_{2.4} (liver: 98 ± 6 s, spleen:

124 ± 19 s), HEM_{1.5} (liver: 108 ± 15 s, spleen: 133 ± 27 s), PEM_{1.5} (liver: 104 ± 9 s, spleen: 148 ± 24 s), indicating that the materials with functional components could efficiently improve control hemorrhage capability (Figure 4a). The hemostasis efficiency in the in vivo rat tail amputation model showed a similar improvement to that of the rat liver and spleen hemostasis model for NEMs (e.g., from 428 ± 42 s (PVPEM) to 190 ± 51 s (KEM_{1.5})). It is worth noting that KEM_{1.5} showed the best hemostasis capability among all the NEMs, which reaches the 'standard of care' level for commercial products (ClG and CoG) in the three in vivo bleeding models (Figure 4a), varied from those reported nanoclay-containing composites^{13,14,16,19}, inorganic nanoparticles³⁹ and other hemostatic materials^{5,6,37,38} that involve poor mechanical strength and inflammation and thrombosis risk due to fine particles or the particles exfoliation from incompact structure (Table S2).

The hemostasis applicability was investigated by observing untreated wounds, treated wounds and wounds after removal of the materials (Figure 4b and Figure S11). There were no residues from the NEMs on the wounds (including a clean wound surface after treated with PVPEM, Figure 4b), in contrast to numerous residues from the hemostatic powders (Figure S11), indicating the practicality and pasting ability of NEMs. And the wound stopped bleeding (liver: 144 ± 22 s, spleen: 227 ± 30 s) treated with PVPEM and was mildly exudative bleeding after 5 minutes again that might due to no active hemostatic components (Figure 4b). Easy use and removal of NEMs play an additional significant role in reducing the risks of wound infection or hemolysis, benefitting from the integrated structures of NEMs. In contrast, the particle residues were extremely difficult to clean and may cause hemolysis or other side effects, which is another major disadvantage of directly using nanoclay for hemostasis (Figure S11 and Supplementary Discussion). The commercial products (ClG and CoG) also exhibited high active component leakage due to poor adhesion, which would necessitate scrupulous debridement⁶.

Supporting Information Page 17 line 11: The active bleeding would not stop over 300 s in blank groups of both spleen and liver wound (the results didn't present in the histogram of Figure S11), and the common medical gauze (the control group of just 'compression') showed a long hemostatic time (liver: 164 ± 56 s, spleen: 201 ± 52 s), which may be due to without any hemostatic active components (Figure S11). There were numerous residues of the three hemostatic powders on the wound that necessitate scrupulous debridement and has a high risk of distal thrombosis and inflammation⁵.

Figure 4. (b) Photographs for the bleeding wound with untreated, treated and removed PVPeM, HEM_{1.5}, PEM_{1.5}, KEM_{1.5}, KEM_{2.0}, KEM_{2.4}, CIG and CoG, respectively, in rat liver and spleen hemostasis model.

Figure S11. (a) Rat liver and spleen hemostasis model. (b) Photographs for the bleeding wound with untreated, treated and removed blank, medical gauze, halloysite, palygorskite, kaolinite group. Blank group hemostasis result didn't present in histogram. (c) Bleeding time of wounds treated with ClG (n = 5), CoG (n = 5 or 6), halloysite (n = 3), palygorskite (n = 3), kaolinite (n = 3) and medical gauze (n = 6 or 5) in rat liver and spleen surface trauma model. All the experiments were replicated at least three times (n ≥ 3) independently with similar results. *P < 0.05, **P < 0.01 using Student's t-test (two-sided). The error bars represent SD. Source data are provided as a Source Data file.

6. Yu, L., Shang, X., Chen, H., Xiao, L., Zhu, Y. & Fan, Jie. A tightly-bonded and flexible mesoporous zeolite-cotton hybrid hemostat. *Nat. Commum* **2019**, *9*, 09849.

16. Feng, C., Li, J., Wu, G. S., Mu, Y. Z., Kong, M., Jiang, C. Q., Cheng, X. J., Liu, Y. & Chen, X. G. Chitosan-Coated Diatom Silica as Hemostatic Agent for Hemorrhage Control. *ACS Appl. Mater. Inter.* **2016**, *8* (50), 34234-34243.

6. In the Material section, the authors should provide more details regarding their experiments. For instance, the parameters used for SEM and DTA characterizations are missing and how the wound condition was scored is not provided.

Reply:

(i) The parameters used for SEM, DTA, TEM or other characterizations have been carefully supplemented in experiment part on Page 16 line 1:

The Diffuse Reflectance Infrared Fourier Transform Spectroscopy (DRIFTS) were tested by a Shimadzu FTIR 8120 spectrometer at a wavenumber range between 4000 and 600 cm⁻¹. The SEM images were examined by a Mira3 LMU SEM (Tescan, Czech Republic) with the voltage of 20 kV. The TEM images and EDX mapping analysis were collected with a Titan G2 60-300 with the voltage of 30 kV. The differential thermal analysis (DTA) was analyzed using the STA449C (NETZSCH, GER), which ranged from 30 to 1000 °C with 5 °C/min rate of rise. XRD measurements were obtained by Rigaku TTR III at range between 5 and 80°. The absorbance (OD) value was obtained from a microplate reader (EnVision Xcite, PerkinElmer). Zeta potentials of samples were measured with a Zeta-sizer Delsa440sx instrument (Beckman Coulter, Brea, CA, USA). The water contact angle was tested with a contact angle goniometer (JY-82C, Chengde Dingsheng Testing Machine Testing Equipment Co., Ltd.).

(ii) The wound conditions were strictly quantified and evaluated to ensure the accuracy of data^{5,16} and the details of wound condition evaluation have been supplemented in experiment part as follows.

Page 17 line 7: All male SD rats were obtained (220~250 g) from the Hunan SJA Laboratory Animal Co., Ltd (Hunan, China), kept in a dedicated feeding box and randomly divided into different groups. All the animal experiments were performed by specific operators for the accuracy of data reducing artificial error, all procedures of which were performed under aseptic condition. All in vivo hemostasis performance evaluations were strictly performed by referencing the relevant standards and literatures^{6,16,42}. The in vivo hemostatic capability of the NEMs were evaluated by representative rat liver and spleen surface trauma model^{6,42} and rat tail amputation model¹⁶.

SD rats were standardly anesthetized with 2% sodium pentobarbital, and then laparotomy was performed to fully expose the liver or spleen in the abdominal cavity⁵. An active bleeding wound of approximately 0.4*0.4 cm on the liver or spleen surface was made by a scalpel. Filter paper was used to make an indicator bar with a width of 0.2 cm, and a black indicator line was drawn 0.5 cm from the end of the indicator bar. After wound bleeding, the indicator bar was immediately placed on the wound. Then, the blood seeped along the indicator bar. If the blood seep to the black line within 3 s, the speed and initial amount of bleeding were qualified for subsequent hemostatic experiments. After completing the bleeding assessment, the same quantity (20 mg) of PVPEM, HEM_{1.5}, PEM_{1.5}, KEM_{1.5}, KEM_{2.0}, KEM_{2.4}, ClG, CoG, halloysite particles, palygorskite particles, kaolinite particles or medical gauze (the control group of just compression) was immediately put on the wound surface with slight compression for the hemostasis experiment. The blank control group only underwent an active bleeding wound without any hemostatic treatment. An indicator bar was used intermittently to assess active bleeding on the wound. If there was no obvious penetration of blood along the indicator bar, there was no active bleeding. Then, the total bleeding time was recorded, and the dress weigh was measured instantly to evaluate blood loss¹⁴.

After the hemostasis experiment, the abdominal cavity was flushed with 10 ml of normal saline^{5,19}. The absorbance value of abdominal cavity lavage fluid was measured

with a microplate reader (EnVision Xcite, PerkinElmer) to indirectly evaluate the amount of bleeding. During the experiment, the heart rate (HR) and blood oxygen saturation (SpO₂) of rats were monitored by a noninvasive animal blood oxygen instrument (DB15, RocSea). After the experiment was completed, the materials and blood clots were removed, and the abdominal cavity was closed. The 5-week survival rate of rats in each group was observed.

Page 18 line 17: All SD rats were evenly and randomly divided into different groups. Then, all SD rats were anesthetized with sodium pentobarbital, and the tail was cut off 5 cm from the end¹⁶. The tail was placed vertically. Blood dripping was observed to ensure a constant speed after cutting, and then subsequent hemostatic experiments were performed. After the bleeding assessment was completed, the tail with active bleeding was immediately placed in a 15 ml centrifuge tube containing the same quantity (100 mg) of PVPEM, HEM_{1.5}, PEM_{1.5}, KEM_{1.5}, KEM_{2.0}, KEM_{2.4}, ClG or CoG to ensure that the wound surface was in contact with the materials with slight compression. The tail was raised intermittently to observe wound bleeding. If the blood volume in the centrifuge tube did not increase significantly, there was no active bleeding due to the completion of hemostasis. Finally, the total time to bleeding stoppage was recorded.

5. Zhao, X., Guo, B., Wu, H., Liang, Y. & Ma, P. X. Injectable antibacterial conductive nanocomposite cryogels with rapid shape recovery for noncompressible hemorrhage and wound healing. *Nat. Commun* **2018**, 9, 2784.

16. Feng, C., Li, J., Wu, G. S., Mu, Y. Z., Kong, M., Jiang, C. Q., Cheng, X. J., Liu, Y. & Chen, X. G. Chitosan-Coated Diatom Silica as Hemostatic Agent for Hemorrhage Control. *ACS Appl. Mater. Inter.* **2016**, 8 (50), 34234-34243.

7. For the histological (Figure 5C) and rat viscera (Figure 5D) analyses, more information should be provided with respect to treatment duration prior to tissue/organ sampling.

Reply:

(i) The detailed description sentences of treatment duration prior to tissue/organ sampling have been supplemented in experiment part on Page 19 line 6:

To evaluate the biocompatibility and degradation performance of the NEMs in vivo, SD rats were anesthetized with sodium pentobarbital, and the hair on the back skin was removed to prepare for surgery⁵. Subcutaneous incisions were formed on the back skin, and materials from each group (20 mg) were embedded under the skin. The blank control group only underwent a subcutaneous incision, which was sutured immediately without embedding any materials. Filter paper (FP) was embedded under the skin to form a positive control group to observe foreign body reactions. The skin reactions of each group were observed on days 0, 3, 7, 14, and 21. After 21 days, the rats were euthanized, and the full-thickness skin tissues of each group were taken for hematoxylin-eosin (HE) staining by a microscopy (Leica, Wetzlar, Germany) to observe the skin pathological changes (especially foreign body reactions and inflammatory reactions) and to evaluate the degradation performance of the materials. In addition, KEM_{1.5} was embedded under the skin alone, and the blank control group was used as a negative control. After 21 days, the rats were euthanized. The heart, lung, stomach, liver, spleen, and kidney were removed to observe the shape and size. Then, the pathological changes in these organs were observed by HE staining to assess whether KEM_{1.5} caused substantial damage to various organs after degradation and absorption. The specific method can be found in our previously published literature¹³.

(ii) The elaborations of the histological and rat viscera analyses have been revised and supplemented in result section on Page 12 line 11:

Images and histological analyses of the injured wounds after the embedding surgery were recorded for 21 days (Figure 5b and c). After 7 days, the wounds produced subcutaneous nodules in the control group, whereas the wounds repaired normally after

implantation with NEMs or when left untreated (blank group). After 21 days, the subcutaneous nodules did not disappear in the control group, while the wounds in the other groups completely recovered with good degradation of NEMs (Figure 5b and c). The histological data were consistent with the results of direct observation (Figure 5c). Severe foreign body granuloma and inflammatory reactions were found in the control group embedding sites, while other samples maintained normal levels of histiocytes (Figure 5c), suggesting that each group of materials (best: KEM_{1.5}) has good biological safety and degradation properties¹³.

Biotoxicity effect of NEMs to other organs was further investigated, including the heart, lung, stomach, liver, spleen, and kidney (Figure 5d), which is necessary and may be influenced by the risk of particle leakage and distal thrombosis⁵. KEM_{1.5} subcutaneous embedding treatment for 21 days showed no significant effect on the shape and size of each organ, which were similar to those of normal rats (Figure 5d). The organs of the rats treated with KEM_{1.5} showed no obvious substantial damage according to histological analysis, indicating no obvious side effects on the circulatory system, respiratory system, digestive system or metabolic system.

5. Zhao, X., Guo, B., Wu, H., Liang, Y. & Ma, P. X. Injectable antibacterial conductive nanocomposite cryogels with rapid shape recovery for noncompressible hemorrhage and wound healing. *Nat. Commum* **2018**, *9*, 2784.

13. Long, M., Zhang, Y., Huang, P., Chang, S., Hu, Y., Yang, Q., Mao, L. & Yang, H. Emerging Nanoclay Composite for Effective Hemostasis. *Adv. Funct. Mater.* **2018**, *28* (10), 1704452.

8. The authors should mention how many replicates and technical replicates have been used in each experiment. In addition, a statistical analysis of their data should be performed, especially in Figures 3, 4, and 5.

Reply: Thanks for reviewer's suggestion. The experiments replicate times (at least three

times for each test with similar results) and the statistical analysis of the data have been supplemented in the corresponding parts in the article or in the legends of Figure 3, 4 and other Figures as follow.

Page 20 line 11: The data in this study are presented as the mean \pm standard deviation (SD). GraphPad Prism (version 8.0) was used for the statistical analysis with Student's t-test between two groups comparison ($*p < 0.05$, $**p < 0.01$, $***p < 0.001$, $****p < 0.0001$). Values of $p < 0.05$ were considered significant. All animal experiments were replicated at least three times independently with similar results, and the differences of exact sample size for each experimental group is associated with the data exclusions from the analyses.

Figure 3c Photographs from the in vitro blood-clotting measurement and the corresponding relative OD values of the supernatant absorbance for (I) PVPPEM, (II) HEM_{1.5}, (III) PEM_{1.5}, (IV) KEM_{1.5}, (V) KEM_{2.0}, (VI) KEM_{2.4}, (VII) ClG, (VIII) CoG, respectively. (n=9) **** $P < 0.0001$ using Student's t-test (two-sided). The error bars represent SD. Source data are provided as a Source Data file.

Figure 4. In vivo clotting research. (a) Bleeding time of wounds treated with PVPeM (n = 7 or 5 or 3), HEM_{1.5} (n = 5 or 4 or 3), PEM_{1.5} (n = 3 or 4), KEM_{1.5} (n = 5 or 3), KEM_{2.0} (n = 4 or 3), KEM_{2.4} (n = 3 or 4), CIG (n = 5 or 4) and CoG (n = 5 or 6 or 3) in rat liver, spleen, rat-tail amputation hemostasis model, respectively. All hemostasis experiments were replicated at least three times (n ≥ 3) independently with similar results. (b) Photographs for the bleeding wound with untreated, treated and removed PVPeM, HEM_{1.5}, PEM_{1.5}, KEM_{1.5}, KEM_{2.0}, KEM_{2.4}, CIG and CoG, respectively, in rat liver and spleen hemostasis model. (c) Abdominal cavity lavage fluid photographs and the corresponding relative OD values for PVPeM (n = 6 or 5), HEM_{1.5} (n = 5 or 4), PEM_{1.5} (n = 3 or 4), KEM_{1.5} (n = 5 or 3), KEM_{2.0} (n = 4), KEM_{2.4} (n = 3 or 4), CIG (n = 5) and CoG (n = 5), respectively, in rat liver and spleen hemostasis model. All the experiments were

replicated at least three times ($n \geq 3$) independently with similar results. $*P < 0.05$, $**P < 0.01$, $***P < 0.001$, $****P < 0.0001$ using Student's t-test (two-sided). The error bars represent SD.

Source data are provided as a Source Data file.

Figure S11c Bleeding time of wounds treated with ClG ($n = 5$), CoG ($n = 5$ or 6), halloysite ($n = 3$), palygorskite ($n = 3$), kaolinite ($n = 3$) and medical gauze ($n = 6$ or 5) in rat liver and spleen surface trauma model. All the experiments were replicated at least three times ($n \geq 3$) independently with similar results. $*P < 0.05$, $**P < 0.01$ using Student's t-test (two-sided). The error bars represent SD. Source data are provided as a Source Data file.

Figure S12. The dressing weigh after (a) liver and (b) spleen hemostasis for PVPEM ($n = 2$ or 3), HEM_{1.5} ($n = 4$), PEM_{1.5} ($n = 3$), KEM_{1.5} ($n = 3$), KEM_{2.0} ($n = 4$), KEM_{2.4} ($n = 3$), ClG ($n = 3$), CoG ($n = 5$ or 6) and medical gauze ($n = 6$ or 5). All the experiments were replicated at least three times ($n \geq 3$) independently with similar results. Source data are provided as a Source Data file.

9. The manuscript needs some editing. There are several misused/misspelled words (e.g. cly instead of clay), and some lengthy sentences need to be revised.

Reply: Thanks for reviewer's suggestion. The errors and misdescriptions have been carefully checked and corrected, and the revised article were further polished and edited by Springer Nature Author Services to improve the flow and readability of the text.

Several representative revise of misspelled words and lengthy sentences:

Page 15 line 9: The electrospun solution was prepared by dispersing nanocly => The electrospun solution was prepared by dispersing nanoclay

Page 15 line 13: dosage of nanocly => dosages of nanoclay were controlled

Page 2 line 18: A practical therapeutic solution for the prevention of acute hemorrhage, especially in military combat and traffic accidents, is urgently needed because of the high clinical demand and large market potential. => A practical prehospital care solution for acute hemorrhage control, especially in the case of warfare, catastrophes and accidents, is urgently needed and is characterized by high clinical demand and large market potential^{2,3}.

Page 5 line 3: As a result, these clays can overcome unnecessary internal linkages (hydrogen bonds) between the clay molecules exposing higher number of functional sites and enhancing the integration with the PVP molecules. => Kaolinite electrospun membranes (KEMs) (best: KEM_{1.5}) exhibit better hemostatic capability in vitro and in vivo with lower biohazardous effects than other nanoclay electrospun membranes (NEMs) and commercial products (ClG and CoG), benefiting from enriched hemostatic functional sites, a robust fluffy framework and a hydrophilic surface.

Page 6 line 15: The magnitude of blueshift indicates that kaolinite has the strongest ability to form hydrogen bonding with PVP compared to that in the case of halloysite and palygorskite; this hydrogen bonding ability becomes stronger concomitant with an increase in the mass ratio of the clay. => The shift increase was concomitant with increased hydrogen bonding interactions in the material. The magnitude of the shift

indicates that kaolinite has a stronger ability than halloysite and palygorskite to form hydrogen bonds with PVP.

Page 12 line 12: PVPEM, HEM_{1.5}, PEM_{1.5}, KEM_{1.5}, KEM_{2.0}, and KEM_{2.4} were implanted into the subcutaneous tissue on the back of rats (Figure 5a) and over the abdomen (Figure 5e) => PVPEM, HEM_{1.5}, PEM_{1.5}, KEM_{1.5}, KEM_{2.0}, and KEM_{2.4} were implanted into subcutaneous tissue on the back of rats (Figure 5a).

Page 13 line 20: In summary, a series of programmable nanoclay-organic self-supported membranes has been developed, and the membranes are directly useable as acute hemostasis bandages. => In summary, a series of designable nanoclay-organic self-supported membranes were developed, which have the potential to be directly used as acute hemostasis bandages.

Page 16 line 2: The structure and an organo-functional group of the as-prepared samples was analyzed by Fourier transform infrared (FTIR) spectroscopy at a wavenumber range between 4000 and 600 cm⁻¹. => The Diffuse Reflectance Infrared Fourier Transform Spectroscopy (DRIFTS) were tested by a Shimadzu FTIR 8120 spectrometer at a wavenumber range between 4000 and 600 cm⁻¹.

Response to Reviewer 2

The authors present the manufacturing of nano clay-impregnated electrospun PVP systems as innovative hemostatic dressings. The preparation/manufacture of the technology and its physico-chemical characterization is very well-described. However, in the context of hemostatic evaluation of the technology, the manuscript is lacking some significant information, experiments and analysis, as follows:

1. The title is very confusing, with the word 'hemostatic' repeated. The 'Rapid Hemostatic' part can be omitted.

Reply: Thanks for the reviewer's professional suggestion. The title has been revised as "Robust Hemostatic Bandages Based on Nanoclay Electrospun Membranes".

2. The authors set up the context of their research in the framework of 'acute hemorrhage is one of the greatest causes of death'. While this is true, this context is actually more for 'non-compressible acute hemorrhage' (internal organ damage, diffuse injury, poly trauma etc.) and not that much for external accessible (often compressible) hemorrhaging injuries. This context should be distinguished in the Intro, especially since the technology that the authors describe is not for non-compressible hemorrhage.

Reply: Thanks for the professional comment on the context, and the distinction on compressible and non-compressible hemorrhage is greatly helpful for improving this study. Acute and unpredictable trauma hemorrhage occurs quickly, and effective hemostasis is critical to relieve life threats instantly¹. The desired hemostatic materials must be designed and developed to achieve rapid definitive hemostasis for both noncompressible (visceral organs) and compressible (cervical, trunk and head regions near the heart and brain) massive bleeding, both of which involve fatal bleeding and are causes of equal concern⁴⁻⁶.

In this study, NEM acute hemostasis bandages with high biosafety were developed, which are excellent and promising candidate materials for **compressible hemorrhage control applications**. The in vitro and in vivo hemostasis performance evaluations were strictly performed by referencing the relevant standards and literatures^{5,6,16}. Specially, there are several common in vivo hemostasis models, including viscera hemostasis model, femoral artery injury model, tail amputation model and tongue incision model. The hemostatic capability of the NEMs were evaluated by representative rat liver and spleen surface trauma model and rat tail amputation model^{6,42}, which was distinguished from the volume defect model of noncompressible hemorrhage hemostasis⁵. The ambiguous description of acute hemorrhage context has been revised and added as follows.

Page 5 line 3: Kaolinite electrospun membranes (KEMs) (best: KEM_{1.5}) exhibit better hemostatic capability in vitro and in vivo with lower biohazardous effects than other nanoclay electrospun membranes (NEMs) and commercial products (CIG and CoG), benefiting from enriched hemostatic functional sites, a robust fluffy framework and a hydrophilic surface. Thus, NEMs exhibit great potential for use as rapid robust hemostats that may be applied as **compressible hemorrhage control** devices.

Page 2 line 18: Acute hemorrhage is one of the greatest causes of death each year around the world¹. A practical prehospital care solution for acute hemorrhage control, especially in the case of warfare, catastrophes and accidents, is urgently needed and is characterized by high clinical demand and large market potential^{2,3}. Rapid effective hemostatic material application in initial hemorrhage phases can lead to an extended rescue time, resulting in a decline in high mortality rates due to excessive bleeding². However, it is still challenging for most hemostatic materials to quickly and safely control hemorrhage from severe bleeding wounds or cuts^{1,4}. The desired hemostatic materials must be designed and developed to achieve rapid definitive hemostasis for both noncompressible (visceral organs) and compressible (cervical, trunk and head regions near the heart and brain) massive bleeding, both of which involve fatal bleeding and are causes of equal concern⁴⁻⁶.

Page 17 line 13: All in vivo hemostasis performance evaluations were strictly performed by referencing the relevant standards and literatures^{6,16,42}. The in vivo hemostatic capability of the NEMs were evaluated by representative rat liver and spleen surface trauma model^{6,42} and rat tail amputation model¹⁶.

1. Baylis, J. R., Ju, H. Y., Thomson, M. H., Kazerooni, A., Xu, W., John, A. E. S., Lim, E. B., Chien, D., Lee, A. & Zhang, J. Q. Self-Propelled Particles that Transport Cargo through Flowing Blood and Halt Hemorrhage. *Sci. Adv.* **2015**, *1* (9), e1500379-e1500379.

4. Shukla, A., Fang, J. C., Puranam, S., Jensen, F. R. & Hammond, P. T. Hemostatic Multilayer Coatings. *Adv. Mater.* **2012**, *24* (4), 492-496.

5. Zhao, X., Guo, B., Wu, H., Liang, Y. & Ma, P. X. Injectable antibacterial conductive nanocomposite cryogels with rapid shape recovery for noncompressible hemorrhage and wound healing. *Nat. Commun* **2018**, *9*, 2784.

6. Yu, L., Shang, X., Chen, H., Xiao, L., Zhu, Y. & Fan, Jie. A tightly-bonded and flexible mesoporous zeolite-cotton hybrid hemostat. *Nat. Commun* **2019**, *9*, 09849.

16. Feng, C., Li, J., Wu, G. S., Mu, Y. Z., Kong, M., Jiang, C. Q., Cheng, X. J., Liu, Y. & Chen, X. G. Chitosan-Coated Diatom Silica as Hemostatic Agent for Hemorrhage Control. *ACS Appl. Mater. Inter.* **2016**, *8* (50), 34234-34243.

42. Pusateri, A. E., McCarthy, S. J., Gregory, K. W., Harris, R. A., Cardenas, L., McManus, A. T. & Goodwin, C. W., Jr. Effect of a chitosan-based hemostatic dressing on blood loss and survival in a model of severe venous hemorrhage and hepatic injury in swine. *J. Trauma* **2003**, *54*, 177-182.

3. Kaolin/Zeolite-embedded hemostat dressings are already available (e.g. Combat Gauze), that have significant hemostatic efficacy. In this framework, it is unclear what advantage or performance enhancement the described electrospun nanoclay-PVP systems are providing.

Reply: Kaolinite based electrospun membranes (KEM) with well-dispersed and partly exposed kaolinite particles on surface displayed the best hemostatic performance (86 ± 10 s in rat liver model and 102 ± 16 s in rat spleen model) compared with previously reported hemostatic materials (Table S2 and Figure 4a). The commercial products QuikClot (zeolite) and Combat Gauze (kaolinite) show effective hemostasis while are limited by inflammation and thrombosis risk due to the particles exfoliated from incompact structure⁶. Nanoclay based hemostatic membranes (NEMs) with integrated structures provide robust framework to anchor active clay components and enhance hemostatic capability with enriched hemostatic functional sites^{6,19}. And the membrane framework could be effectively supported by nanoclay fillers that could resist the

spontaneous shrinkage of PVP electrospun membrane^{34,35} (e.g., KEM_{1.5}, < 30% shrinkage ratio), providing a large effective area and a high active component utilization.

Representative nanoclay based and other hemostasis materials were summarized in Table S2, and the detailed descriptions of hemostatic advantage of NEMs have been revised and supplemented as follows.

Page 3 line 22: A few types of biomaterials using nanoclays have been developed as effective hemostats, including hydrogels (e.g., gelatin/polyacrylamide (PAAm)/laponite hydrogels¹⁸) and sponges (e.g., graphene-kaolinite or graphene-montmorillonite composite sponge^{14,19}), which are strict and complicated treatments with high costs as medical hemostats. Hydrogels and sponges show good shape memory but poor mechanical strength due to their inherent interconnected macroporous structures, and polymer binders fill and block the clay pores and layer gaps, causing low clay utilization as active components^{6,16}.

Page 10 line 10: It showed a long hemostatic time for the control group (common medical gauze: liver, 164 ± 56 s; spleen, 201 ± 52 s, Figure S11) and PVPEM group (liver: 144 ± 22 s, spleen: 227 ± 30 s). The bleeding times showed clear decreases with NEM treatment: KEM_{1.5} (liver: 86 ± 10 s, spleen: 102 ± 16 s), KEM_{2.0} (liver: 108 ± 15 s, spleen: 118 ± 16 s) and KEM_{2.4} (liver: 98 ± 6 s, spleen: 124 ± 19 s), HEM_{1.5} (liver: 108 ± 15 s, spleen: 133 ± 27 s), PEM_{1.5} (liver: 104 ± 9 s, spleen: 148 ± 24 s), indicating that the materials with functional components could efficiently improve control hemorrhage capability (Figure 4a). The hemostasis efficiency in the in vivo rat tail amputation model showed a similar improvement to that of the rat liver and spleen hemostasis model for NEMs (e.g., from 428 ± 42 s (PVPEM) to 190 ± 51 s (KEM_{1.5})). It is worth noting that KEM_{1.5} showed the best hemostasis capability among all the NEMs, which reaches the 'standard of care' level for commercial products (ClG and CoG) in the three in vivo bleeding models (Figure 4a), varied from those reported nanoclay-containing composites^{13,14,16,19}, inorganic nanoparticles³⁹ and other hemostatic materials^{5,6,37,38} that involve poor mechanical strength and inflammation and

thrombosis risk due to fine particles or the particles exfoliation from incompact structure (Table S2).

Page 13 line 21: In KEM, sheet-like kaolinite was uniformly dispersed in and partly exposed on PVP fibers, facilitating the formation of a stabilizing framework and regulating spontaneous shrinkage and degradation of PVP fibers; these properties were superior to those of PVPEM, HEM and PEM. KEM_{1.5}, with its enriched hemostatic functional sites, robust framework and hydrophilic surface, has better comprehensive performances on hemostatic time, hemostatic effect and blood loss compared with those of the other NEMs, CIG and CoG in an in vivo rat tail amputation model and a rat liver and spleen injury model. KEM (best: KEM_{1.5}) with a highly hydrophilic framework promotes blood coagulation by rapidly absorbing plasma and aggregating blood components. These results indicate that KEM acute hemostasis bandages with high biosafety are excellent and promising candidate materials for compressible hemorrhage control applications.

Figure 4. In vivo clotting research. (a) Bleeding time of wounds treated with PVPEM (n = 7 or 5 or 3), HEM_{1.5} (n = 5 or 4 or 3), PEM_{1.5} (n = 3 or 4), KEM_{1.5} (n = 5 or 3), KEM_{2.0} (n = 4 or 3), KEM_{2.4} (n = 3 or 4), CIG (n = 5 or 4) and CoG (n = 5 or 6 or 3) in rat liver, spleen, rat-tail amputation hemostasis model, respectively. All hemostasis experiments were replicated at least three times (n ≥ 3) independently with similar results.

Table S2 Representative examples of various materials recently reported for rapid hemostasis applications

Hemostasis materials and the main components		Benefits	Remarks (specific characteristics, applications and limitations)	Reference
Commercially available products	QuikClot (zeolite) and Combat Gauze (kaolinite)	High porosity, large specific surface area and good biocompatibility facilitating to the blood coagulation through absorbing plasma components as well as easily for accessibility and applicability	Fast hemostatic performance but the exothermicity (for zeolite) and the poor adhesion active component and substrate existing the risk of residual and difficulty for removal from the wounds as well as restriction on internal tissue hemorrhage application	--
Other uncommercialized materials	N-Alkyl chitosan nanofiber membrane	Activation of coagulation factors and platelets for blood coagulation in vitro	Good effects for minor bleeding but the poor mechanical properties and adhesivity unsuitable for severe bleeding	1
	Self-assembling peptides (RADA16-I)	Physical entrapment of blood components in a network of peptide	Excellent performance for hemostasis but the complex fabrication process and expensive cost restriction on practical application	2
	CNT-reinforced glycidyl methacrylate (GMA) functionalized quaternized chitosan (QCSG/CNT)	Injectable and conductive shape memory cryogels presents excellent hemostatic performance in rabbit liver defect lethal noncompressible hemorrhage model (Time: ~3.9 min)	Suitable for irregularly shaped hemorrhage but the residual CNT existing the risk of thrombus and difficulty for removal from wound after bleeding-stop	3
	Mesoporous silica nanoparticles (MSN)	Promoting the blood proteins to contact the huge interior surfaces of MSN and then initiate the quick blood clot	Accessibility and diffusion of clotting-promoting proteins for hemostasis but the fine MSN particles existing the risk of thrombus and histogenic immunity	4
	Mesoporous zeolite-cotton hybrid hemostat (mCHA-C)	Mesoporous zeolite onto the surface of cotton fiber by on-site template-free growth with higher procoagulant activity and minimized loss of active components in rabbit lethal femoral artery injury model (Time: 159 ± 12 s)	Great effects for severe hemorrhage but the complicated fabrication process restricted practical application and existing risk of secondary damage on the wound from exothermicity	5
	Gelatin/PAAm/Laponite Nanocomposite Hydrogels	Hierarchical structures, high transparencies, good homogeneity and ultrastrong tensibilities	Less toxic effects but restriction on severe hemolysis, poor mechanical strength and long clotting time	6
	Graphene-montmorillonite or graphene-kaolin composite sponge (GMCS)	Quickly absorbing plasma to increase the concentration of hemocytes and platelets in rabbit artery injury test (Time: ~85 s)	Good biocompatibility but the exfoliated and residual MMT or kaolin particles difficulty for removal and degradation from bleeding tissue site	7,8
This work	Integrated Electrospun Membranes Nanoclay (best: KEM _{1.5})	Accelerate the coagulation of blood through rapidly absorbing plasma and aggregating blood components with a highly hydrophilic framework, and FXII could be activated by kaolinite on the integrated fiber surface to trigger the intrinsic coagulation cascade	The NEMs with the integrated structure and robust framework have excellent hemostatic capability with low biotoxicity	--

6. Yu, L., Shang, X., Chen, H., Xiao, L., Zhu, Y. & Fan, Jie. A tightly-bonded and flexible mesoporous zeolite-cotton hybrid hemostat. *Nat. Commum* **2019**, *9*, 09849.

19. Liang, Y., Xua, C., Lia, G., Liub, T., Liangb, J. F. & Wang X. Graphene-kaolin composite sponge for rapid and riskless hemostasis. *Colloid Surface B* **2018**, *169*, 168-175.

34. Suganya, S., Ram, T. S., Lakshmi, B. S. & Giridev, V. R. Herbal drug incorporated antibacterial nanofibrous mat fabricated by electrospinning: an excellent matrix for wound dressings. *J. Appl. Polym. Sci* **2011**, *121*, 2893-2899.

35. Jin, Y., Hwang, S., Ha, H., Park, H., Kang, S.-W., Hyun, S., Jeon, S. & Jeong, S.-H. Buckled Au@PVP nanofiber networks for highly transparent and stretchable conductors. *Adv. Electron. Mater.* **2016**, *2*, 1500302.

4. In the in vivo evaluation of hemostatic efficacy, while the models are well-described (and well-known) models of bleeding, the comparison groups do not include any commercially available well-established 'standard of care' (e.g. QuikClot or Combat Gauze), and also does not include a control group of just 'compression'. These are necessary to demonstrate/establish what hemostatic advantage do the described nanoclay-PVP materials provide beyond 'standard of care' commercial products that are already out there. If such 'standard of care' are not available in the country/location of the reported research, then it should be clearly stated.

Reply: The hemostatic performance was supplemented and investigated for comparisons of NEMs, the 'standard of care' commercial products (QuikClot ClG and CoG) and the control group of just 'compression' (common medical gauze). The common medical gauze showed a long hemostatic time (liver: 164 ± 56 s, spleen: 201 ± 52 s), while the NEMs (best for KEM_{1.5}: liver, 86 ± 10 s; spleen, 102 ± 16 s) showed obvious improvement of hemostasis efficiency, which reaches the 'standard of care'

level for commercial products (CIG and CoG) (Figure 4a and Figure S11). And KEM_{1.5} treatment showed the lightest dress weigh (33 ± 6 mg) than that of medical gauze (42 ± 10 mg), CIG (40 ± 10 mg) and CoG (46 ± 11 mg) and the lowest abdominal cavity lavage fluid OD value after hemostasis in viscera hemostasis model (Figure S12 and Figure 4c), indicating less blood loss and a faster procoagulant capability for KEM_{1.5}³⁷⁻³⁹. The descriptions of in vivo evaluation of hemostatic efficacy have been revised and added in the corresponding parts as follows.

Page 10 line 8: When active bleeding wounds (the initial bleeding was assessed to ensure a same test condition, Figure S10) of the liver and spleen occurred, the bleeding time was more than 300 s without hemostatic material treatment. It showed a long hemostatic time for the control group (common medical gauze: liver, 164 ± 56 s; spleen, 201 ± 52 s, Figure S11) and PVPEM group (liver: 144 ± 22 s, spleen: 227 ± 30 s). The bleeding times showed clear decreases with NEM treatment: KEM_{1.5} (liver: 86 ± 10 s, spleen: 102 ± 16 s), KEM_{2.0} (liver: 108 ± 15 s, spleen: 118 ± 16 s) and KEM_{2.4} (liver: 98 ± 6 s, spleen: 124 ± 19 s), HEM_{1.5} (liver: 108 ± 15 s, spleen: 133 ± 27 s), PEM_{1.5} (liver: 104 ± 9 s, spleen: 148 ± 24 s), indicating that the materials with functional components could efficiently improve control hemorrhage capability (Figure 4a). The hemostasis efficiency in the in vivo rat tail amputation model showed a similar improvement to that of the rat liver and spleen hemostasis model for NEMs (e.g., from 428 ± 42 s (PVPEM) to 190 ± 51 s (KEM_{1.5})). It is worth noting that KEM_{1.5} showed the best hemostasis capability among all the NEMs, which reaches the 'standard of care' level for commercial products (CIG and CoG) in the three in vivo bleeding models (Figure 4a), varied from those reported nanoclay-containing composites^{13,14,16,19}, inorganic nanoparticles³⁹ and other hemostatic materials^{5,6,37,38} that involve poor mechanical strength and inflammation and thrombosis risk due to fine particles or the particles exfoliation from incompact structure (Table S2).

Page 11 line 16: Blood loss, which is a key index for hemostasis capability, was investigated by assessing the dressing weight after hemostasis¹⁹. In the liver model (Figure S12), the KEM_{1.5} dressing weight after hemostasis was 33 ± 6 mg (initial

sample weight: 20 mg), which was lighter than that of the commercial products (ClG: 40 ± 10 mg, CoG: 46 ± 11 mg). The dressing weights in the spleen model showed a similar result to those in the liver model (lightest: KEM_{1.5}), and the KEM_{1.5} treatment showed the lowest abdominal cavity lavage fluid relative OD value after hemostasis from liver and spleen wounds, which means less blood loss and a faster procoagulant capability for KEM_{1.5} (Figure 4c).

Figure 4. In vivo clotting research. (a) Bleeding time of wounds treated with PVPEM (n = 7 or 5 or 3), HEM_{1.5} (n = 5 or 4 or 3), PEM_{1.5} (n = 3 or 4), KEM_{1.5} (n = 5 or 3), KEM_{2.0} (n = 4 or 3), KEM_{2.4} (n = 3 or 4), ClG (n = 5 or 4) and CoG (n = 5 or 6 or 3) in rat liver, spleen, rat-tail amputation hemostasis model, respectively. All hemostasis experiments were replicated at least three times (n

≥ 3) independently with similar results. (b) Photographs for the bleeding wound with untreated, treated and removed PVPEM, HEM_{1.5}, PEM_{1.5}, KEM_{1.5}, KEM_{2.0}, KEM_{2.4}, ClG and CoG, respectively, in rat liver and spleen hemostasis model. (c) Abdominal cavity lavage fluid photographs and the corresponding relative OD values for PVPEM (n = 6 or 5), HEM_{1.5} (n = 5 or 4), PEM_{1.5} (n = 3 or 4), KEM_{1.5} (n = 5 or 3), KEM_{2.0} (n = 4), KEM_{2.4} (n = 3 or 4), ClG (n = 5) and CoG (n = 5), respectively, in rat liver and spleen hemostasis model. All the experiments were replicated at least three times (n ≥ 3) independently with similar results. **P* < 0.05, ***P* < 0.01, ****P* < 0.001, *****P* < 0.0001 using Student's t-test (two-sided). The error bars represent SD. Source data are provided as a Source Data file.

Figure S11. (a) Rat liver and spleen hemostasis model. (b) Photographs for the bleeding wound with untreated, treated and removed blank, medical gauze, halloysite, palygorskite, kaolinite group. Blank group hemostasis result didn't present in histogram. (c) Bleeding time of wounds treated with

ClG (n = 5), CoG (n = 5 or 6), halloysite (n = 3), palygorskite (n = 3), kaolinite (n = 3) and medical gauze (n = 6 or 5) in rat liver and spleen surface trauma model. All the experiments were replicated at least three times (n ≥ 3) independently with similar results. *P < 0.05, **P < 0.01 using Student's t-test (two-sided). The error bars represent SD. Source data are provided as a Source Data file.

Figure S12. The dressing weigh after (a) liver and (b) spleen hemostasis for PVPEM (n = 2 or 3), HEM_{1.5} (n = 4), PEM_{1.5} (n = 3), KEM_{1.5} (n = 3), KEM_{2.0} (n = 4), KEM_{2.4} (n = 3), ClG (n = 3), CoG (n = 5 or 6) and medical gauze (n = 6 or 5). All the experiments were replicated at least three times (n ≥ 3) independently with similar results. Source data are provided as a Source Data file.

37. Wang, X., Guan, J., Zhuang, X., Li, Z., Huang, S., Yang, J., Liu, C., Li, F., Tian, F., Wu, J. & Shu Z. Exploration of blood coagulation of N-Alkyl chitosan nanofiber membrane in vitro. *Biomacromolecules*. **2018**, *19*, 731-739.

38. Hsu, B. B., Conway, W., Tschabrunn, C. M., Mehta, M., Perez-Cuevas, M. B., Zhang, S. & Hammond P. T. Clotting mimicry from robust hemostatic bandages based on self-assembling peptides. *ACS Nano* **2015**, *9*, 9394–9406.

39. Chen, Z., Li, F., Liu, C., Guan, J., Hu, X., Du, G., Yao, X., Wu, J. & Tian, F. Blood clot initiation by mesoporous silica nanoparticles: dependence on pore size or particle size. *J. Mater. Chem. B* **2013**, *1*, 1-10.

5. The only 'quantitative data' provided is 'hemostats time' and there is no statistical analysis/difference described between any of the groups compared. For hemostatic analysis, one should additionally analyze 'blood loss' (e.g. by weighing the dressing) and also analyze the 'clotting mechanisms' (e.g. cellular and molecular components of the clotting process. like platelet activation, thrombin generation, fibrin content, hemolytic aspects etc.).

Reply:

(i) The statistical analysis of the data has been supplemented in experiment part and in Figures 3, 4 and other Figures as follows.

Page 20 line 11: The data in this study are presented as the mean \pm standard deviation (SD). GraphPad Prism (version 8.0) was used for the statistical analysis with Student's t-test between two groups comparison ($*p < 0.05$, $**p < 0.01$, $***p < 0.001$, $****p < 0.0001$). Values of $p < 0.05$ were considered significant. All animal experiments were replicated at least three times independently with similar results, and the differences of exact sample size for each experimental group is associated with the data exclusions from the analyses.

Figure 3c Photographs from the in vitro blood-clotting measurement and the corresponding relative OD values of the supernatant absorbance for (I) PVPPEM, (II) HEM_{1.5}, (III) PEM_{1.5}, (IV) KEM_{1.5}, (V) KEM_{2.0}, (VI) KEM_{2.4}, (VII) ClG, (VIII) CoG, respectively. (n=9) $****P < 0.0001$ using

Student's t-test (two-sided). The error bars represent SD. Source data are provided as a Source Data file.

Figure 4a Bleeding time of wounds treated with PVPEM (n = 7 or 5 or 3), HEM_{1.5} (n = 5 or 4 or 3), PEM_{1.5} (n = 3 or 4), KEM_{1.5} (n = 5 or 3), KEM_{2.0} (n = 4 or 3), KEM_{2.4} (n = 3 or 4), CIG (n = 5 or 4) and CoG (n = 5 or 6 or 3) in rat liver, spleen, rat-tail amputation hemostasis model, respectively. All hemostasis experiments were replicated at least three times (n ≥ 3) independently with similar results.

Figure S11c Bleeding time of wounds treated with CIG (n = 5), CoG (n = 5 or 6), halloysite (n = 3), palygorskite (n = 3), kaolinite (n = 3) and medical gauze (n = 6 or 5) in rat liver and spleen surface trauma model. All the experiments were replicated at least three times (n ≥ 3) independently with similar results. *P < 0.05, **P < 0.01 using Student's t-test (two-sided). The error bars represent SD. Source data are provided as a Source Data file.

(ii) The evaluation of blood loss was supplemented by referencing the related literatures^{5,19}. The bleeding wound with KEM_{1.5} treatment showed the lightest dress weigh (33 ± 6 mg) than that of medical gauze (42 ± 10 mg), ClG (40 ± 10 mg) and CoG (46 ± 11 mg) and the lowest abdominal cavity lavage fluid OD value after hemostasis in viscera hemostasis model (Figure S12 and Figure 4c), indicating less blood loss and a faster procoagulant capability for KEM_{1.5}. The coagulation mechanism in vitro involved full and easy contact with the active particles on NEMs (best: KEM_{1.5}), promoting blood component aggregation and accelerating red blood cell activation (Figure 3d and 3e). Furthermore, the intrinsic (contact activation) coagulation cascade pathway was activated by KEM_{1.5} contact to stimulate the conversion of FXII to FXIIa, generating thrombin and fibrin and finally forming blood clots¹³ (Figure S15).

More description sentences of blood loss and clotting mechanisms were added and revised as follows.

Page 9 line 19: The hemostatic mechanism in vitro was further investigated by observing the morphology of blood cells components adhered to KEM_{1.5} surfaces (Figure 3d). The blood components could easily permeate and interact with partly exposed kaolinite particles after blood was dropped on the KEM_{1.5} surface, promoting a quick clot. Red blood cells aggregated at the contact point with KEM_{1.5}, showing activated states with deformed shapes⁵. Thus, the coagulation mechanism in vitro involved full and easy contact with the active particles on KEM_{1.5}, promoting blood component aggregation and accelerating red blood cell activation (illustration in Figure 3e).

Page 11 line 16: Blood loss, which is a key index for hemostasis capability, was investigated by assessing the dressing weight after hemostasis¹⁹. In the liver model (Figure S12), the KEM_{1.5} dressing weight after hemostasis was 33 ± 6 mg (initial sample weight: 20 mg), which was lighter than that of the commercial products (ClG: 40 ± 10 mg, CoG: 46 ± 11 mg). The dressing weights in the spleen model showed a

similar result to those in the liver model (lightest: KEM_{1.5}), and the KEM_{1.5} treatment showed the lowest abdominal cavity lavage fluid relative OD value after hemostasis from liver and spleen wounds, which means less blood loss and a faster procoagulant capability for KEM_{1.5} (Figure 4c).

Page 13 line 5: Furthermore, coagulation related tests were investigated to explore the effect of NEMs on clotting cascade activation. The TT, FIB, PT and APTT of all the NEMs were within 15.07-17.33 s, 2.81-3.29 g/L, 8.47-11.37 s and 25.17-35.46 s (Figure S14 and Supplementary Discussion), respectively, all among normal ranges. The NEMs showed no significant effect on TT, FIB, PT, or APTT, which indirectly suggested no obvious bleeding or thrombosis risk for local and other surrounding organs, indicating that the hemostatic mechanism of NEMs may not be significantly related to extrinsic (tissue factor) coagulation cascade pathway activation⁶. As previous work and discussion¹³, it could be that the intrinsic (contact activation) coagulation cascade pathway was activated by KEM_{1.5} contact to stimulate the conversion of FXII to FXIIa, generating thrombin and fibrin and finally forming blood clots (Figure S15).

Figure 3d The SEM images of the interfacial interactions between blood cells and KEM_{1.5}. (e) Illustration of blood permeation to the KEM_{1.5} and the interactions between red blood cells and KEM_{1.5}.

Figure 4. (b) Photographs for the bleeding wound with untreated, treated and removed PVPEM, HEM_{1.5}, PEM_{1.5}, KEM_{1.5}, KEM_{2.0}, KEM_{2.4}, CIG and CoG, respectively, in rat liver and spleen hemostasis model. **(c)** Abdominal cavity lavage fluid photographs and the corresponding relative OD values for PVPEM (n = 6 or 5), HEM_{1.5} (n = 5 or 4), PEM_{1.5} (n = 3 or 4), KEM_{1.5} (n = 5 or 3), KEM_{2.0} (n = 4), KEM_{2.4} (n = 3 or 4), CIG (n = 5) and CoG (n = 5), respectively, in rat liver and spleen hemostasis model. All the experiments were replicated at least three times (n ≥ 3) independently with similar results. **P* < 0.05, ***P* < 0.01, ****P* < 0.001, *****P* < 0.0001 using Student's t-test (two-sided). The error bars represent SD. Source data are provided as a Source Data file.

Figure S12. The dressing weight after (a) liver and (b) spleen hemostasis for PVPEM (n = 2 or 3), HEM_{1.5} (n = 4), PEM_{1.5} (n = 3), KEM_{1.5} (n = 3), KEM_{2.0} (n = 4), KEM_{2.4} (n = 3), CIG (n = 3), CoG (n = 5 or 6) and medical gauze (n = 6 or 5). All the experiments were replicated at least three times (n ≥ 3) independently with similar results. Source data are provided as a Source Data file.

Figure S15. Illustration of the intrinsic (contact activation) coagulation cascade pathway activated by KEM in hemostasis.

5. Zhao, X., Guo, B., Wu, H., Liang, Y. & Ma, P. X. Injectable antibacterial conductive nanocomposite cryogels with rapid shape recovery for noncompressible hemorrhage and wound healing. *Nat. Commun* **2018**, *9*, 2784.

13. Long, M., Zhang, Y., Huang, P., Chang, S., Hu, Y., Yang, Q., Mao, L. & Yang, H. Emerging Nanoclay Composite for Effective Hemostasis. *Adv. Funct. Mater.* **2018**, *28* (10), 1704452.

19. Liang, Y., Xua, C., Lia, G., Liub, T., Liangb, J. F. & Wang X. Graphene-kaolin composite sponge for rapid and riskless hemostasis. *Colloid Surface B* **2018**, *169*, 168-175.

6. The impact of a hemostatic technology is not only in 'reducing blood loss' but resultantly stabilizing vitals and improving survival time. It is unclear if any such measurement and analysis was done in the in vivo model. Ideally, if it is a heavy bleeding model from liver and/or spleen, one should observe rapid destabilization of mean arterial pressure, SpO₂ etc., and with hemostat stabilization these aspects would be 'rescued'. One should also observe and record 'lethality' (survival), if the model is a lethal model. None of this was reported in the manuscript and so it is unclear what the hemostatic impact actually is.

Reply: The changes in rat heart rate and blood oxygen saturation (SpO₂) were evaluated and supplemented during the in vivo hemostatic process¹³. The destabilized rat heart rate after laparotomy and trauma treatments showed a rapid rescued and stabilization treated with NEMs accompanied by stable blood oxygen saturation (SpO₂) (Figure S13). For the blank group, the wound was given compression to stop the bleeding after 5 minutes due to severe destabilization of heart rate and SpO₂ and unceasing active bleeding that may be related to no active hemostatic components. With continuous

observation to hemostasis completely, there was no significant effect on the heart rate and SpO₂ of the rats treated with NEMs (best: KEM_{1.5}) treatment. After the hemostasis experiment was done, the abdominal cavity was closed and a 100% survival of rats was retained over 5 weeks observation in each group (Figure S13), confirming good hemostasis and rescue effects of NEMs (best: KEM_{1.5})³⁸.

The detailed evaluation descriptions of rescue and hemostatic performances have been added in result part and supplementary discussion part as follows.

Page 11 line 23: In addition, the destabilized rat heart rate after laparotomy and trauma treatments showed a rapid rescued and stabilization treated with NEMs accompanied by stable blood oxygen saturation (SpO₂), which provided a statistically significant advantage for retaining a 100% survival rate over 5 weeks of observation after hemostasis (Figure S13 and Supplementary Discussion), confirming good hemostasis and rescue effects³⁸.

Supporting Information, Page 17 line 17: The changes in rat heart rate and blood oxygen saturation (SpO₂) were evaluated during the hemostatic process of each material (Figure S13). The destabilized rat heart rate after laparotomy and trauma treatments showed a rapid rescued and stabilization treated with NEMs accompanied by stable blood oxygen saturation (SpO₂). For the blank group, the wound was given compression to stop the bleeding after 5 minutes due to severe destabilization of heart rate and SpO₂ and unceasing active bleeding that may be related to no active hemostatic components. Under continuous observation to hemostasis completely, there was no significant effect on the heart rate and blood oxygen saturation of the rats treated with NEMs (best: KEM_{1.5}), suggesting that the blood loss was not enough to cause death or obvious hemorrhagic shock. After the hemostasis experiment was done, the abdominal cavity was closed and a 100% survival of rats was retained over 5 weeks observation in each group (Figure S13).

Figure S13. (a) The heart rates, (b) blood oxygen saturations, (c) survival rates of PVPEM, HEM_{1.5}, PEM_{1.5}, KEM_{1.5}, KEM_{2.0}, KEM_{2.4} CoG, CoG and blank group in rat liver hemostasis model. (d) The heart rates, (e) blood oxygen saturations, (f) survival rates of PVPEM, HEM_{1.5}, PEM_{1.5}, KEM_{1.5}, KEM_{2.0}, KEM_{2.4} CoG, CoG and blank group in rat spleen hemostasis model. For heart rate and blood oxygen saturation assessment before laparotomy, after laparotomy and after spleen bleeding, the data of all rats were collected to assess the overall changes. Source data are provided as a Source Data file.

13. Long, M., Zhang, Y., Huang, P., Chang, S., Hu, Y., Yang, Q., Mao, L. & Yang, H. Emerging Nanoclay Composite for Effective Hemostasis. *Adv. Funct. Mater.* **2018**, 28 (10), 1704452.

38. Hsu, B. B., Conway, W., Tschabrunn, C. M., Mehta, M., Perez-Cuevas, M. B., Zhang, S. & Hammond P. T. Clotting mimicry from robust hemostatic bandages based on self-assembling peptides. *ACS Nano* **2015**, 9, 9394–9406.

7. Since this is a localized (and not systemic) application, the rationale behind testing multi-organ toxicity is confusing. Testing locally the inflammatory aspects, as well as the potential of exothermic reactions (if any) from the clay on the tissue should be the focus of the tests.

Reply: Thanks for reviewer's suggestion. The local inflammatory effect of NEMs was investigated by images and histological analyses (Figure 5b and c)⁵. Severe foreign body granuloma and inflammatory reactions were found in the control group (filter paper) embedding sites after 21 days embedding observation, while the NEMs groups maintained normal levels of histiocytes (Figure 5c). Furthermore, it is necessary to evaluate the toxic effect to surrounding organs that may be influenced by the risk of particle leakage and distal thrombosis⁶. The rats treated with KEM_{1.5} showed no obvious substantial damage and inflammatory on other organs according to histological analysis (Figure 5d), which was consistent with direct observation results, indicating good biocompatible effects on the circulatory system, respiratory system, digestive system or metabolic system. It was worth noting that there was no exothermicity of clays¹⁹, and thus all results indicated good biological safety and degradation property of NEMs (best: KEM_{1.5}).

The ambiguous descriptions of biosafety evaluation have been revised and supplemented as follows.

Page 12 line 8: To further confirm the biosafety of the monolithic materials, PVPEM, HEM_{1.5}, PEM_{1.5}, KEM_{1.5}, KEM_{2.0}, and KEM_{2.4} were implanted into subcutaneous tissue on the back of rats (Figure 5a), and the tests included a control group (filter paper) and a blank group. Images and histological analyses of the injured wounds after the embedding surgery were recorded for 21 days (Figure 5b and c). After 7 days, the wounds produced subcutaneous nodules in the control group, whereas the wounds repaired normally after implantation with NEMs or when left untreated (blank group). After 21 days, the subcutaneous nodules did not disappear in the control group, while the wounds in the other groups completely recovered with good degradation of NEMs (Figure 5b and c). The histological data were consistent with the results of direct observation (Figure 5c). Severe foreign body granuloma and inflammatory reactions were found in the control group embedding sites, while other samples maintained

normal levels of histiocytes (Figure 5c), suggesting that each group of materials (best: KEM_{1.5}) has good biological safety and degradation properties¹³.

Page 12 line 21: Biotoxicity effect of NEMs to other organs was further investigated, including the heart, lung, stomach, liver, spleen, and kidney (Figure 5d), which is necessary and may be influenced by the risk of particle leakage and distal thrombosis⁵. KEM_{1.5} subcutaneous embedding treatment for 21 days showed no significant effect on the shape and size of each organ, which were similar to those of normal rats (Figure 5d). The organs of the rats treated with KEM_{1.5} showed no obvious substantial damage according to histological analysis, indicating no obvious side effects on the circulatory system, respiratory system, digestive system or metabolic system.

Figure 5. Biological toxicity evaluation. (a) Photograph for the wound regions in back embedding model. (b) Photographs of the wounds that were embedded the PVPEM, HEM_{1.5}, PEM_{1.5}, KEM_{1.5}, KEM_{2.0}, KEM_{2.4}, control group and blank group samples immediately following surgery (day 0), and for 3, 7, 14, 21 day. (c) Histological analysis of subcutaneous tissue of the wounds after embedding surgery. (d) Photographs and histological analysis for rat internal organs (heart, lung,

stomach, liver, spleen and kidney) after 21 days embedding experiment with KEM_{1.5} and blank group, respectively.

5. Zhao, X., Guo, B., Wu, H., Liang, Y. & Ma, P. X. Injectable antibacterial conductive nanocomposite cryogels with rapid shape recovery for noncompressible hemorrhage and wound healing. *Nat. Commun* **2018**, *9*, 2784.

6. Yu, L., Shang, X., Chen, H., Xiao, L., Zhu, Y. & Fan, Jie. A tightly-bonded and flexible mesoporous zeolite-cotton hybrid hemostat. *Nat. Commun* **2019**, *9*, 09849.

19. Liang, Y., Xua, C., Lia, G., Liub, T., Liangb, J. F. & Wang X. Graphene-kaolin composite sponge for rapid and riskless hemostasis. *Colloid Surface B* **2018**, *169*, 168-175.

REVIEWER COMMENTS

Reviewer #2 (Remarks to the Author):

The authors have presented significant extent of revised information and data in their manuscript revision, that has addressed majority of the critique comments. This is very commendable. A few specific and important aspects remain un-answered in the revision, and the publication of the manuscript with addressing (at least discussing) these aspects will be very impactful. These are as follows:

1. In vitro characterization of the 'mechanism' of action of the KEM membrane systems: Figure 3 and Figure S9 results to characterize 'blood coagulation' are somewhat rudimentary/observational and not mechanistic. The RBC shape change aspect is an 'effect' of the coagulation occurring, possibly due to biomechanical forces rendered by fibrin on RBCs, and not a 'mechanistic cause' of coagulation. Authors have used the term 'RBC... activated states with deformed shapes' and it is unclear what that means. More appropriate tests would have been Platelet Activation analysis, PT/aPTT, Thrombin Generation Assay, SEM of fibrin morphology etc., that are more reflective of 'coagulation mechanism' rather than 'coagulation effect'.

2. There is no statistical difference in any of the important 'hemorrhage control/hemostasis' effects between KEM, control groups as well as comparison groups (CIG and CoG) in the in vivo models, as indicated by Fig S13 and S14, whereas the in vivo data shown in Fig. 4 among the same groups show differential in 'relative OD' value as well as 'bleeding time' value for the various injury locations. This is a bit confusing and possibly deserves some discussion and explanation in the manuscript.

Other than these two aspects, the bulk of the revisions are adequate and have addressed the previous review critique.

RESPONSES TO REVIEWERS

Reviewer #2 (Remarks to the Author):

The authors have presented significant extent of revised information and data in their manuscript revision, that has addressed majority of the critique comments. This is very commendable. A few specific and important aspects remain un-answered in the revision, and the publication of the manuscript with addressing (at least discussing) these aspects will be very impactful. These are as follows:

1. In vitro characterization of the 'mechanism' of action of the KEM membrane systems: Figure 3 and Figure S9 results to characterize 'blood coagulation' are somewhat rudimentary/observational and not mechanistic. The RBC shape change aspect is an 'effect' of the coagulation occurring, possibly due to biomechanical forces rendered by fibrin on RBCs, and not a 'mechanistic cause' of coagulation. Authors have used the term 'RBC... activated states with deformed shapes' and it is unclear what that means. More appropriate tests would have been Platelet Activation analysis, PT/aPTT, Thrombin Generation Assay, SEM of fibrin morphology etc., that are more reflective of 'coagulation mechanism' rather than 'coagulation effect'.

Reply: Thanks for your professional and valuable suggestions on Figure 3 and Figure S9 results. Furtherly, the KEM coagulation mechanism was analyzed by Scanning Electron Microscope (SEM) images, Flow Cytometry (FCM) and Thromboelastogram. Figure 3d (surface) and Figure S10 (interior) shown that KEM_{1.5} with 2D multilayered structure and 3D reticular architecture facilitated to restrict blood flow and capture blood components. In FCM analysis (Figure S11)^{37,38}, the percentage of CD61⁺ cells in KEM_{1.5} group (0.49%), ClG group (2.92%), CoG group (2.55%) showed an obvious

decrease compared to blank group (93.48%), due to the free platelet counts decrease caused by hemostats capture^{37,39,40}. Importantly, the percentage of CD61⁺/CD62P⁺ cells in KEM_{1.5} group (80.28%) was highest among all groups (blank: 24.68%; CIG: 74.98%; CoG: 71.43%), indicating that KEM_{1.5} have an excellent platelet activation effect³⁹.

In Thromboelastogram analysis (Figure S12)^{41,42}, KEM_{1.5} group (Coagulation composite index CI: 1.30; Reaction time R: 1.6 min) exhibited a hypercoagulability with high clotting factors activity compared to blank group (CI: -8.07; R: 11.80 min). As previous work and the related literatures^{42,13,19}, the whole blood contacts with the kaolinite particles will activate kininogen, plasma kallikrein, and factor XII to trigger the process of coagulation. All the values of Maximum amplitude (MA), Clot formation time (K), Angle, Peripheral blood platelets-monocyte aggregates (PMA), Prediction of fibrinolysis index (EPL), Percentage lysis 30 min post-MA (LY30) and Mechanical strength of clot (G) were among the normal ranges in KEM_{1.5} group, maintaining a general-level functions on platelets, fibrinogen, fibrinolysis and clot strength.

The revised description of Figure 3 and S9 and the detailed analysis of coagulation mechanism have been supplemented as follows.

Page 9 line 6: The results clearly showed that KEM and the commercial controls (CoG and CIG) could facilitate blood permeation and coagulation within 2~5 min (best for KEM_{1.5}, < 2 min). => The observational results clearly showed that KEM and the commercial controls (CoG and CIG) could facilitate blood permeation and coagulation within 2~5 min (best for KEM_{1.5}, < 2 min).

Page 10 line 15: Thus, the coagulation mechanism in vitro involved full and easy contact with the active particles on KEM_{1.5}, promoting blood component aggregation and accelerating red blood cell activation (illustration in Figure 3e). => Thus, a full and easy contact with the active particles on KEM_{1.5} can accelerate the blood components aggregation (illustration in Figure 3e) and the contact activation (intrinsic) process of clotting (including factor XII, thrombin and fibrin formation until the final clot, illustration in Figure S13).

Page 9 line 20: The KEM coagulation mechanism was further analyzed by Scanning Electron Microscope (SEM) images, Flow Cytometry (FCM) and Thromboelastogram. Figure 3d (surface) and Figure S10 (interior) shown that KEM_{1.5} with 2D multilayered structure and 3D reticular architecture facilitated to restrict blood flow and capture blood components. In FCM analysis (Figure S11)^{37,38}, the percentage of CD61⁺ cells in KEM_{1.5} group (0.49%), ClG group (2.92%), CoG group (2.55%) showed an obvious decrease compared to blank group (93.48%), due to the free platelet counts decrease caused by hemostats capture^{37,39,40}. Importantly, the percentage of CD61⁺/CD62P⁺ cells in KEM_{1.5} group (80.28%) was highest among all groups (blank: 24.68%; ClG: 74.98%; CoG: 71.43%), indicating that KEM_{1.5} have an excellent platelet activation effect³⁹.

In Thromboelastogram analysis (Figure S12)⁴¹⁻⁴³, KEM_{1.5} group (Coagulation composite index CI: 1.30; Reaction time R: 1.6 min) exhibited a hypercoagulability with high clotting factors activity compared to blank group (CI: -8.07; R: 11.80 min). As previous work and the related literatures^{42,13,19}, the whole blood contacts with the kaolinite particles will activate kininogen, plasma kallikrein, and factor XII to trigger the process of coagulation. All the values of Maximum amplitude (MA), Clot formation time (K), Angle, Peripheral blood platelets-monocyte aggregates (PMA), Prediction of fibrinolysis index (EPL), Percentage lysis 30 min post-MA (LY30) and Mechanical strength of clot (G) were among the normal ranges in KEM_{1.5} group, maintaining a general-level functions on platelets, fibrinogen, fibrinolysis and clot strength.

Page 17 line 23 (Methods): 100 μ L whole blood was dropped onto 10 mg membrane sample for each group. The samples were added with 10 μ L 0.2 M CaCl₂ solutions immediately and incubated at 37 °C for 5 min. The prepared samples were chemically fixed using 2.5% glutaraldehyde in PBS (pH=7.0) at 25 °C for 4 h. Then, the samples were dehydrated with PBS (two times), 25%, 50%, 75%, 80%, 90%, and 100% (two times) ethanol serially (treated 15 min for each step). The dried samples were

investigated the blood cells contact and aggregation (surface and interior) by SEM^{16,46}.

5 μ L fresh whole blood with sodium citrate (9:1) anticoagulant was dropped on 10 mg NEMs membrane sample for each group (incubation for 10 min). 100 μ L PBS, 2 μ L antibody CD61 (the platelet markers) and 2 μ L antibody CD62P (the dominant platelet activation markers)^{37,38} were added and bound to the samples then oscillated to fully disperse (incubation for 15 min). The samples were diluted using 200 μ L PBS then filtrated the NEMs samples using a filter. The percentage of CD61⁺ and double CD61⁺/CD62P⁺ cells was analyzed by a flow cytometer (DxP Athena V2-B4-R2, Wuxi Xiatai Biological Technology Co., Ltd.). 100 mg sample was added to 1 mL whole blood with sodium citrate (9:1) anticoagulant for each group (incubation for 10 min). 340 μ L prepared whole blood solutions were calcified with 20 μ L 0.2 M CaCl₂ solution in the test cups. Then, the samples were investigated using a Thromboelastogram analyzer (CFMS LEPU-8800, Lepu Medical Technology (Beijing) Co., Ltd.)^{41,42}.

Figure S10. The SEM images of blood cells in the 3D reticular architecture of KEM_{1.5}. Red

represents red blood cells; Yellow represents the platelets.

Figure S11. Platelet aggregation and activation: The percentage of CD61⁺ and CD61⁺/CD62P⁺ cells after function with KEM_{1.5}, CIG, CoG and without materials (blank group), respectively. * $P < 0.05$, ** $P < 0.01$, *** $P < 0.001$, **** $P < 0.0001$ using Student's t-test (two-sided); $n=4$. Source data are provided as a Source Data file.

Figure S12. The values and traces of Thrombelastograph (TEG) of the whole blood samples after function with KEM_{1.5}, CoG, CoG and without materials (blank group), respectively. (CI= Coagulation composite index, R=Reaction time, MA= Maximum amplitude, K=Clot formation time, Angle = α° of the greatest amplitude on the TEG trace, G= Mechanical strength of clot, PMA= Peripheral blood platelets-monocyte aggregates, EPL= Prediction of fibrinolysis index, LY30= Percentage lysis 30 min post-MA.). * $P < 0.05$, ** $P < 0.01$ using Student's t-test (two-sided); $n=3$. Source data are provided as a Source Data file.

Figure 3. Shrinkage and coagulation properties of nanoclay electrospun membranes in vitro. (d) The SEM images of blood cells on the KEM_{1.5} surface.

5. Zhao, X., Guo, B., Wu, H., Liang, Y. & Ma, P. X. Injectable antibacterial conductive nanocomposite cryogels with rapid shape recovery for noncompressible hemorrhage and wound

healing. *Nat. Commun.* **2018**, *9*, 2784.

6. Yu, L., Shang, X., Chen, H., Xiao, L., Zhu, Y. & Fan, J. A tightly-bonded and flexible mesoporous zeolite-cotton hybrid hemostat. *Nat. Commun.* **2019**, *9*, 09849.

13. Long, M., Zhang, Y., Huang, P., Chang, S., Hu, Y., Yang, Q., Mao, L. & Yang, H. Emerging nanoclay composite for effective hemostasis. *Adv. Funct. Mater.* **2018**, *28* (10), 1704452.

19. Liang, Y., Xua, C., Lia, G., Liub, T., Liangb, J. F. & Wang X. Graphene-kaolin composite sponge for rapid and riskless hemostasis. *Colloid Surface B* **2018**, *169*, 168-175.

37. Blair, T. A. & Frelinger III, A. L. Platelet surface marker analysis by mass cytometry. *Platelets.* **2019**, 1369-1635.

38. Dave, R. G., Geevar, T., Aaron, S., Benjamin, R. N., Mammen, J., Kumar, S., Vijayan, R., Gowri, M. & Nair, S. C. Diagnostic utility of flow cytometry based coated-platelets assay as a biomarker to predict thrombotic or hemorrhagic phenotype in acute stroke. *Cytom. Part B-Clin. Cy.* **2021**, 22026.

39. Velzen, J. F. v., Gorkom, B. A.P. L., Pop, G. A. M. & Heerde, W. L. v. Multicolor flow cytometry for evaluation of platelet surface antigens and activation markers. *Thromb. Res.* **2012**, *130*, 92-98.

40. Ortiz-Rivero, S., Baquero, C., Hernández-Cano, L., Roldán-Etcheverry, J. J., Gutiérrez-Herrero, S., Fernández-Infante, C., Martín-Granado, V., Anguita, E., Pereda, J. M. d., Porras, A. & Guerrero C. C3G, through its GEF activity, induces megakaryocytic differentiation and proplatelet formation. *Cell Commun. Signal.* **2018**, *16*, 101.

41. Differding, J. A., Underwood, S. J., Van, P. Y., Khaki, R. A., Spoerke, N. J. & Schreiber, M. A. Trauma induces a hypercoagulable state that is resistant to hypothermia as measured by thrombelastogram. *Am. J. Surg.* **2011**, *201*, 587-591.

42. Oliver, W. C. *Anticoagulation and Coagulation Management for ECMO* (Sage, 2009, 154-157).

43. Spiezia, L., Marchioro, P., Radu, C., Rossetto, V., Tognin, G., Monica, C., Salmaso L. & Simioni, P. Whole blood coagulation assessment using rotation thrombelastogram thromboelastometry in patients with acute deep vein thrombosis. *Blood Coagul. Fibrin.* **2008**, *19*, 355-360.

2. There is no statistical difference in any of the important 'hemorrhage control/hemostasis' effects between KEM, control groups as well as comparison groups (ClG and CoG) in the in vivo models, as indicated by Fig S13 and S14, whereas the in vivo data shown in Fig. 4 among the same groups show differential in 'relative OD' value as well as 'bleeding time' value for the various injury locations. This is a bit confusing and possibly deserves some discussion and explanation in the manuscript.

Reply: Thank you for the careful and considerate comments on Figure S17 and S18 (S13 and S14 in original version), and the statistical difference between KEM_{1.5}, blank group and comparison groups have been supplemented. The rat heart rate showed a significant difference ($P < 0.0001$) before and after laparotomy and trauma treatments, which was attributed to the destabilization caused by bleeding (Figure S17). Under continuous observation to hemostasis completion, all NEMs treatment groups (including ClG and CoG) showed a rapid rescue and stabilization for the destabilized rat heart rate accompanied by stable blood oxygen saturation (SpO₂). Although in the liver hemorrhage model, there was a statistical difference in SpO₂ between before laparotomy and KEM_{1.5} treatment, the difference is very small and does not have actual clinical significance. It emphasized that there were no significant differences between KEM_{1.5} and comparison groups on the heart rate and SpO₂, which was consistent with the results in bleeding time and 'relative OD' value, indicating that KEM_{1.5} reaches the 'standard of care' level for commercial products (ClG and CoG)^{6,46}. After the hemostasis experiment was done (including an additional compression to stop the bleeding after 5 minutes in blank group), the abdominal cavity was closed and a same (100%) survival of rats was retained over 5 weeks observation in each group (Figure S17).

The TT, FIB, PT and APTT of all the NEMs, ClG and CoG were within 15.07-17.33 s, 2.81-3.29 g/L, 8.47-11.37 s and 25.17-35.46 s (including INR (0.88-0.99) and PTA (95.7-118.2%)), respectively (Figure S18). Although those results showed slight differences between some groups, the results of TT, FIB, PT, or APTT in all groups were all among normal ranges. Therefore, in the original version of this figure, it wasn't specifically emphasized statistical differences. It also means that a general level was

maintained on TT, FIB, PT, or APTT for all NEMs, CIG and CoG, suggesting a good safety of coagulation system^{6,45}.

The revised Figures and detailed descriptions have been added in result part and supplementary discussion part, as follows.

Figure S17. (a) The heart rates, (b) blood oxygen saturations, (c) survival rates of PVPEM, HEM_{1.5}, PEM_{1.5}, KEM_{1.5}, KEM_{2.0}, KEM_{2.4}, CIG, CoG and blank group in rat liver hemostasis model. (d) The heart rates, (e) blood oxygen saturations, (f) survival rates of PVPEM, HEM_{1.5}, PEM_{1.5}, KEM_{1.5}, KEM_{2.0}, KEM_{2.4}, CIG, CoG and blank group in rat spleen hemostasis model. For heart rate and blood oxygen saturation assessment before laparotomy, after laparotomy and after spleen bleeding, the data of all rats were collected to assess the overall changes. All the experiments were replicated at least three times ($n \geq 3$) independently with similar results. * $P < 0.05$, ** $P < 0.01$, *** $P < 0.001$, **** $P < 0.0001$ using Student's t-test (two-sided). Source data are provided as a Source Data file.

Figure S18. (a) The thrombin time (TT), (b) fibrinogen (FIB), (c) prothrombin time (PT), (d) activated partial thromboplastin time (APTT), (e) international Normalized Ratio (INR), (f) prothrombin activity (PTA) for PVPEM, HEM_{1.5}, PEM_{1.5}, KEM_{1.5}, KEM_{2.0}, KEM_{2.4} ClG and CoG, respectively. All the experiments were replicated at least three times ($n \geq 3$) independently with similar results. $*P < 0.05$, $**P < 0.01$, $***P < 0.001$, $****P < 0.0001$ using Student's t-test (two-sided). Source data are provided as a Source Data file.

Page 13 line 19: Furthermore, coagulation related tests were investigated to explore the effect of NEMs on clotting cascade, concerning the safety of coagulation system. The TT, FIB, PT and APTT of all the NEMs were within 15.07-17.33 s, 2.81-3.29 g/L, 8.47-11.37 s and 25.17-35.46 s (Figure S18 and Supplementary Discussion), respectively, all among normal ranges. It means that a general level was maintained on TT, FIB, PT, or APTT treated with NEMs, which indirectly suggested no obvious bleeding or thrombosis risk for local and other surrounding organs, indicating a good safety of coagulation system^{6,45}. And it indicated that the hemostatic mechanism of NEMs may not be significantly related to extrinsic (tissue factor) coagulation cascade pathway activation^{13,45}, which is consistent with the results of Thromboelastogram (Figure S12).

Supporting Information Page 20 line 18: The changes in rat heart rate and blood oxygen saturation (SpO₂) were evaluated during the hemostatic process of each material (Figure S17). The rat heart rate showed a significant difference ($P < 0.0001$) before and after

laparotomy and trauma treatments, which was attributed to the destabilization caused by bleeding (Figure S17). Under continuous observation to hemostasis completion, all NEMs treatment groups (including CIG and CoG) exhibited a rapid rescue and stabilization for the destabilized rat heart rate accompanied by stable blood oxygen saturation (SpO₂). For the blank group, the wound was given compression to stop the bleeding after 5 minutes due to severe destabilization of heart rate and SpO₂ and unceasing active bleeding that may be related to no active hemostatic components. Although in the liver hemorrhage model, there was a statistical difference in SpO₂ between before laparotomy and KEM_{1.5} treatment, the difference is very small and does not have actual clinical significance. It emphasized that there were no significant differences between KEM_{1.5} and comparison groups on the heart rate and SpO₂, which was consistent with the results in bleeding time and 'relative OD' value, indicating that KEM_{1.5} reaches the 'standard of care' level for commercial products (CIG and CoG). After the hemostasis experiment was done (including an addition compression to stop the bleeding after 5 minutes in blank group), the abdominal cavity was closed and a same (100%) survival of rats was retained over 5 weeks observation in each group (Figure S17).

Furthermore, coagulation related tests were investigated to explore the effect of NEMs on clotting cascade, concerning the safety of coagulation system. The TT, FIB, PT and APTT of all the NEMs, CIG and CoG were within 15.07-17.33 s, 2.81-3.29 g/L, 8.47-11.37 s and 25.17-35.46 s (including INR (0.88-0.99) and PTA (95.7-118.2%)), respectively (Figure S18). Although those results showed slight differences between some groups, the results of TT, FIB, PT, or APTT in all groups were all among normal ranges¹⁴. It also means that a general level was maintained on TT, FIB, PT, or APTT for all NEMs, CIG and CoG, suggesting a good safety of coagulation system.

6. Yu, L., Shang, X., Chen, H., Xiao, L., Zhu, Y. & Fan, J. A tightly-bonded and flexible mesoporous zeolite-cotton hybrid hemostat. *Nat. Commun.* **2019**, 9, 09849.

13. Long, M., Zhang, Y., Huang, P., Chang, S., Hu, Y., Yang, Q., Mao, L. & Yang, H. Emerging nanoclay composite for effective hemostasis. *Adv. Funct. Mater.* **2018**, 28 (10), 1704452.

45. Wang, X., Guan, J., Zhuang, X., Li, Z., Huang, S., Yang, J., Liu, C., Li, F., Tian, F., Wu, J. & Shu Z. Exploration of blood coagulation of N-Alkyl chitosan nanofiber membrane in vitro. *Biomacromolecules.* **2018**, 19, 731-739.

46. Hsu, B. B., Conway, W., Tschabrunn, C. M., Mehta, M., Perez-Cuevas, M. B., Zhang, S. & Hammond P. T. Clotting mimicry from robust hemostatic bandages based on self-assembling peptides. *ACS Nano* **2015**, 9, 9394-9406.

REVIEWERS' COMMENTS

Reviewer #2 (Remarks to the Author):

The authors have adequately addressed my comments with additional data, discussion and rationale. I do not have any further question. In my review, this revised manuscript is appropriate for acceptance.

RESPONSE TO REVIEWERS

Reviewer #2 (Remarks to the Author):

The authors have adequately addressed my comments with additional data, discussion, and rationale. I do not have any further question. In my review, this revised manuscript is appropriate for acceptance.

Reply: I take this opportunity to extend my heartfelt appreciation to the kind assistance you render me during the manuscript processing.